

# Chemical and meteorological influences on the lifetime of NO₃ at a semi-rural mountain site during "PARADE"

N. Sobanski[1], M.J. Tang[1,5], J. Thieser[1], G. Schuster[1], D. Pöhler[2], H. Fischer[1], W. Song[1], C. Sauvage[1], J. Williams[1], J. Fachinger[3], F. Berkes[4,6], P. Hoor[4], U. Platt[2], J. Lelieveld[1] and J.N. Crowley[1]

[1] Max-Planck-Institut für Chemie, Division of Atmospheric Chemistry, Mainz, Germany.
[2] Institute of Environmental Physics, University of Heidelberg, Germany.
[3] Max-Planck-Institut für Chemie, Division of Particle Chemistry, Mainz, Germany.
[4] Institut For Atmospheric Physics, Johannes Gutenberg-University Mainz, Germany.

[5] Present affiliation: Chemistry Department, University of Iowa, Iowa, USA.
[6] Present affiliation: Institute of Energy and Climate, Forschungszentrum Jülich, Jülich, Germany.

*Correspondence to*: J. N. Crowley (john.crowley@mpic.de)

**Abstract.** Through measurements of NO₂, O₃, and NO₃ during the PARADE campaign (PArticles and RAdicals, Diel observations of mEchanisms of oxidation) in the German Taunus mountains we derive nighttime, steady state lifetimes ($\tau_{ss}$) of NO₃ and N₂O₅. During some nights, high NO₃ (~200 pptv) and N₂O₅ (~1 ppbv) mixing ratios are associated with values of $\tau_{ss}$ that exceeded one hour for NO₃ and three hours for N₂O₅ near the ground. Such long boundary layer lifetimes for NO₃ and N₂O₅ are usually only encountered in very clean/unreactive air masses whereas the PARADE measurement site is impacted by both biogenic emissions from the surrounding forest and anthropogenic emissions from the nearby urbanized/industrialised centres. Measurement of several trace gases which are reactive towards NO₃ indicate that the inferred lifetimes are significantly longer than those calculated from the summed loss rate. Several potential causes for the apparently extended NO₃ and N₂O₅ lifetimes are examined, including additional routes to formation of NO₃ and the presence of a low-lying residual layer. Overall, the most likely cause of the anomalous lifetimes are related to the meteorological conditions, though additional NO₃ formation due to reactions of Criegee intermediates may contribute.

## 1 Introduction

NO₃ and N₂O₅ are key species in the chemical removal of NOx and of several hydrocarbons at night (Wayne et al., 1991; Atkinson and Arey, 2003). Traditionally, NO₃ is considered to be formed mainly by the reaction of NO₂ with O₃ (R1) with negligible contributions from e.g. reaction between OH and HNO₃ (R2) or the photolysis of halogen nitrates (XONO₂), where X may be e.g. Br or Cl (R3). N₂O₅ is produced in an association reaction between NO₃ and NO₂ (R4), with which it is in thermal equilibrium (R4, R-4).

$$NO_2 + O_3 \quad \rightarrow \quad NO_3 + O_2 \tag{R1}$$



| OH + HNO$_3$ | $\rightarrow$ | NO$_3$ + H$_2$O | (R2) |
| XONO$_2$ + $h\nu$ | $\rightarrow$ | NO$_3$ + X | (R3) |
| NO$_2$ + NO$_3$ + M | $\rightarrow$ | N$_2$O$_5$ + M | (R4) |
| N$_2$O$_5$ + M | $\rightarrow$ | NO$_3$ + NO$_2$ + M | (R-4) |

The production rate of NO$_3$ is therefore usually assumed to be the product of the NO$_2$ and O$_3$ concentrations and the rate constant ($k_1$):

$$P(NO_3) = k_1[NO_2][O_3] \qquad \text{(Eq1)}$$

In general, equilibrium between N$_2$O$_5$, NO$_3$ and NO$_2$ is reached in a few minutes following sunset and holds until sunrise

(Brown et al., 2003). During the day, NO$_3$ is rapidly photolysed (with a lifetime of a few seconds) to give NO$_2$ + O (R5a, ~90 %) or NO + O$_2$ (R5b, ~10%) (Johnston et al., 1996) and also reacts quickly with NO (R6) so that NO$_3$ and N$_2$O$_5$ mixing ratios are usually below the detection limit of most instruments. During the night, in the absence of light to regenerate it from NO$_2$ (R7) and with sufficient distance from emissions sources, NO levels approach zero (due to R8) and NO$_3$ and N$_2$O$_5$ can accumulate in the atmosphere.

| NO$_3$ + $h\nu$ | $\rightarrow$ | NO$_2$ + O | (R5a) |
| NO$_3$ + $h\nu$ | $\rightarrow$ | NO + O$_2$ | (R5b) |
| NO$_3$ + NO | $\rightarrow$ | 2 NO$_2$ | (R6) |
| NO$_2$ + $h\nu$ | $\rightarrow$ | NO + O | (R7) |
| NO + O$_3$ | $\rightarrow$ | NO$_2$ + O$_2$ | (R8) |

The mechanism and rate of removal of NO$_3$ and N$_2$O$_5$ from the atmosphere depends on the type of air mass. In areas impacted by local anthropogenic activity, NO can be a major sink of NO$_3$. The rate constant for this reaction of $2.6 \times 10^{-11}$ cm$^3$ molecule$^{-1}$ s$^{-1}$ at 298 K (Atkinson et al., 2004) results, for example, in a chemical lifetime for NO$_3$ of about 16 s in the presence of 100 pptv of NO. NO$_3$ mixing ratios measured in-situ at ground level in cities are sometimes low due to the high mixing ratio of NO. However, a few tens of kilometres downwind of urban areas or several tens of meters aloft, where [NO]

can be significantly lower due to reduced mixing of ground level emissions and titration by O$_3$, NO$_3$ mixing ratios can reach up to several hundred pptv and N$_2$O$_5$ up to a few ppbv (Asaf et al., 2009; Brown et al., 2009). Downwind of industrial activity, especially that related to petroleum production and storage, observed NO$_3$ lifetimes are limited by the presence of reactive hydrocarbons (Geyer et al., 2003; Stutz et al., 2010; Brown et al., 2011; Crowley et al., 2011).

In rural and forested environments, NO can be emitted from soils but is in general much less concentrated than in urban

areas, and its mixing ratio is often close to zero at nighttime. In these environments, biogenic volatile organic compounds (BVOCs) can contribute a substantial fraction of the NO$_3$ reactivity. Unsaturated BVOCs such as monoterpenes have large rate constants for reaction with NO$_3$ e.g. ~10$^{-11}$ cm$^3$ molecule$^{-1}$ s$^{-1}$ for limonene and $10^{-12}$ cm$^3$ molecule$^{-1}$ s$^{-1}$ for α-pinene and β-pinene (Atkinson and Arey, 2003). In heavily forested areas with high monoterpene emission rates, NO$_3$ lifetimes may be




reduced to a few tens of seconds (Rinne et al., 2012; Mogensen et al., 2015) resulting in efficient conversion of NO$x$ to organic nitrates and formation of secondary organic aerosol (Hallquist et al., 1999; Fry et al., 2014).

In the absence of any known, significant gas-phase reaction, the main sink of $N_2O_5$ at night is heterogeneous hydrolysis on aqueous aerosols, clouds and surfaces. The formation of particle phase $HNO_3$ by this route results in repartitioning of NO$x$

from the gas to the condensed phase. Deposition of the particle nitrate formed is a substantial, permanent loss process for NO$x$, resulting in reduced rates of photochemical $O_3$ production. Furthermore, if the aerosol particles contain chloride, the uptake of $N_2O_5$ can result in release of $ClNO_2$ and subsequently Cl atoms, which may enhance the rates of oxidation of hydrocarbons (Behnke et al., 1997; Osthoff et al., 2008; Bertram and Thornton, 2009; Thornton et al., 2010; Sarwar et al., 2012). The $N_2O_5$ uptake coefficient (overall efficiency of transfer from gas to particle phase) is largest for aqueous aerosols

but depends on the composition of the aerosol (nitrate, sulphate and organic content) and humidity and possible organic coatings (Riemer et al., 2009) and is thus highly variable in time and space. Losses of $NO_3$ to aerosol or other surfaces are generally considered to be minor compared to its gas-phase losses and indirect losses via $N_2O_5$ uptake. In environments such as clean marine air masses where reactive partners for $NO_3$ are absent, lifetimes can become large and $N_2O_5$ losses to aerosol or surfaces may control the $NO_3$ levels via the equilibrium (R4, R-4) (Heintz et al., 1996; Martinez et al., 2000).

Over the continents during the night, when the earth's surface does not receive energy by solar radiation, it cools down and the convective mixing which usually takes place during the day stops. This results in formation of a shallow, weakly mixed nocturnal boundary layer (NBL), with strong vertical stratification. Under such conditions, pronounced vertical gradients in $NO_3$ and $N_2O_5$ mixing ratios have been observed (von Friedeburg et al., 2002; Stutz et al., 2004; Brown et al., 2007) with ground level emissions (both biogenic and anthropogenic) often resulting in lower $NO_3$ and $N_2O_5$ lifetimes, with cleaner air

aloft (e.g. in the residual layer or free troposphere) often resulting in higher values. In all these different environments, the emission/formation rates of VOCs and aerosols and subsequent mixing are thus the key parameters controlling the $NO_3$ and $N_2O_5$ lifetimes.

In this paper, we describe a subset of an extensive suite of measurements obtained in the PARADE campaign (2011) at the top of a ~825 m high mountain situated between a urbanized flatland area and a forested mountainous area. A first campaign

at this site (Crowley et al., 2010) during May 2009 found maximum $NO_3$ lifetimes of 10 to 15 minutes with mixing ratios up to 50 pptv for $NO_3$ and 500 pptv for $N_2O_5$. From the results of this first campaign it was concluded that at near-zero nighttime NO mixing ratios, the $NO_3$ lifetime was most likely controlled by emissions of BVOCs, which were however not measured. The heterogeneous loss of $N_2O_5$ was found to be only of limited importance in controlling $NO_3$ lifetimes. During PARADE, several hydrocarbons, both biogenic and anthropogenic were measured as well as NO and also aerosol surface

area and chemical composition. We compare $NO_3$ lifetimes calculated using a steady-state approximation (based on the $NO_3$ production rate and mixing ratio) with those derived from the summed reactivity of individual trace gases and surfaces. For most campaign nights, we found that the steady state lifetime significantly exceeded the lower limit of the $NO_3$ lifetime calculated from the summed reactivity and investigate potential causes of this.



## 2 Campaign location and measurement techniques

### 2.1 Site description

The PARADE campaign took place at the "Taunus Observatory" located on top of the "Kleiner Feldberg" mountain (50.22 N, 8.45 E) 825 m above sea level (ASL). The station is used permanently by the environment and geological agency of the
state of Hessen (HLUG) and the German weather service (DWD). The site has already been described (Handisides, 2001; Crowley et al., 2010), and only a short summary is given below. Directly (1 – 2 km) to the North-North-East and the South-East of the site are two similarly high mountains, ("Großer Feldberg" 878 m and "Altkoenig" 798 m ASL). The area in the North-West to North-East sector may be described as a relatively sparsely populated, partially forested, rural region, whereas the South-West to South-East sector contains significant urban infrastructure including the densely populated cites of
Frankfurt (~ 20 km to the South-East) and Mainz and Wiesbaden (20 – 30 km to the South-West). The site is directly surrounded (100 m radius) by a mix of coniferous trees (mainly spruce) and shrubs. Within a radius of 5 km from the site, the region is dominated by forest (coniferous, broad-leafed and mixed), especially to the NE which has the lowest contribution from local agriculture and urban emissions. This is displayed in the land-use pie charts found in Fig. S1 of the supplementary information. The same figure indicates that over a distance of 50 km, a significant fraction (~ 50 %) of the
land area is used for agriculture.

The largest urban influence is found in the South-East sector for both 5 and 50 km distances to the site. In the northern sectors, the closest cites are located between 60 and 90 km away from the site and are significantly less populated than Frankfurt, Mainz and Wiesbaden. The heavily populated and industrialized "Ruhr Gebiet" is about 130 km towards the North-West.

During the PARADE campaign most instruments were located in the permanent laboratory facilities at the site. A platform on top of the building was used to house some instruments and the inlets outside at a height (from the ground) of about 8 m. Measurements central to this study (NO, $NO_2$, $NO_3$, $N_2O_5$, $O_3$, VOCs) were sampled via collocated inlets (< 5 m horizontal separation, < 1 m vertical separation). Aerosol measurements were made from a mobile laboratory "MoLa" (Drewnick et al., 2012) with its inlet at 10 m height, about 15 m distant from the main building. The instruments deployed are listed in
Table 1, operational details, measurement uncertainties and detection limits are given in the next section. The MoLa also provided meteorological data (Wind-speed, wind-direction, temperature, pressure relative humidity and global radiation) at 1 s resolution. The values were in generally good agreement with the lower frequency datasets reported by the DWD and HLUG.

### 2.2 Instrumentation

Below, we describe the instruments (and their overall uncertainties) used to make the critical measurements related to the calculation of $NO_3$ lifetimes. $NO_2$ and $O_3$, important for calculation of $NO_3$ production rates were measured by multiple instruments, with excellent agreement. A comparison of three $NO_2$ datasets, including the one used here, is described in





(Thieser et al., 2015). Several $O_3$ datasets agreed to within about 10 % and the choice of dataset for e.g. $NO_2$ or $O_3$ would not change the conclusions of this work.

### 2.2.1 $NO_2$, $NO_3$ and $N_2O_5$ measurements by CRDS

The two-channel, thermal dissociation cavity-ringdown-spectrometer (hereafter CRDS) measuring $NO_3$ and $N_2O_5$ was

deployed at this site during a previous campaign (Schuster et al., 2009; Crowley et al., 2010). Briefly, the CRDS uses a laser diode tuned to 662 nm to directly measure $NO_3$ in one channel at ambient temperature and the sum of ($NO_3$ + $N_2O_5$) in a second channel which is held at ~ 100 °C, converting $N_2O_5$ to $NO_3$ (plus $NO_2$). The mixing ratio of $N_2O_5$ is then calculated from the difference in mixing ratios observed in each cavity. The CRDS instrument was located outside on the platform on top of the building in order to reduce inlet loss of $NO_3$. Air was drawn through a 1 m length of ½" (OD) PFA tube at 50 (std)

Lmin$^{-1}$ (SLM) and sampled (18 SLM) from the centre of the flow via ¼" PFA tubing into the two cavities of the CRDS. This setup keeps inlet residence times short (≈ 0.1 s) and also helps avoid sampling of coarse particles and droplets. Corrections were applied to account for the losses of $NO_3$ on the filters, on the walls of the automatic filter-changer and during passage through the cavities. The overall $NO_3$ transmission was ~ 68 % (Crowley et al., 2010). The instrument detection limit was 2 pptv in 1 s for $NO_3$ and approximately 5 pptv for $N_2O_5$. The total uncertainty was ~15 % for both $NO_3$ and $N_2O_5$, with the

largest contributions arising from uncertainty in the $NO_3$ cross section and $NO_3$ losses.

During PARADE, $NO_2$ was measured using several instruments. We use data obtained using CRDS with a 405 nm diode laser as recently described in detail (Thieser et al., 2015). The instrument has a detection limit for $NO_2$ of ≈ 20 pptv in 1 min and an accuracy of 6 %.

### 2.2.2 VOC measurement by GC.

VOCs were measured using two gas-chromatographic instruments with mass spectrometer (GC-MS) and flame ionisation detectors (GC-FID). The GC-MS measured biogenic and aromatic hydrocarbons with on-line adsorption/thermal desorption (Markes International) connected to a gas chromatograph (Agilent GC 6890A) and a mass selective detector (Agilent MSD 5973 inert). The sampling time was 35min, detection limits were around 1 ppt with an uncertainty of 10-15 %.

The GC-FID measured non-methane hydrocarbons (NMHC) using two coupled gas chromatographs (GC 5000 VOC; GC

5000 BTX, AMA instruments, Ulm Germany). GC 5000 VOC was used for measurement of C2-C6 NMHCs and BTX for C6-C12. The detection limits ranged from 1-5 ppt, exceptions being ethane, ethane, propene, benzene and toluene with values of 8 ppt, 16 ppt, 9 ppt, 14 ppt, 48 ppt, respectively. The GC-FID was calibrated using a multi-gas mixture (National Physical Laboratory, 2010). Total uncertainty is close to 10 % for most trace-gases with the exception of 1-pentane (15 %). The time resolution of the quasi online measurement is 60 min and the mixing ratio represents an average over a sampling

period of 20 min.



### 2.2.3 NO, O₃ and CO₂

NO measurements were made with a modified, commercial Chemiluminescence Detector (CLD 790 SR). Operation of this instrument during the PARADE campaign has recently been described (Li et al., 2015). The detection limit for this instrument is 4 pptv in 2 s with a total uncertainty of 4%. The $O_3$ dataset used in this work was obtained by a commercial

AirPointer from the Recordum Messtechnik GmbH company. It was located in the MoLa, and has a detection limit of ~ 1 ppbv (Drewnick et al., 2012) and an accuracy of 5 %. The $O_3$ inlet was situated a few meters above the MoLa, itself located 15 m meters away from the main tower. $CO_2$ was measured using non-dispersive differential broadbanding Infrared-absorption (LICOR 6262 ).

### 2.2.4 NO₂, O₃ and NO₃ measurements by LP-DOAS

In addition to the co-located, point measurements described above, $NO_2$, $O_3$ and $NO_3$ were also measured by long-path differential absorption spectroscopy (LP-DOAS) over a light path of ~3 km. The telescope was situated on the same roof as the CRDS instruments (835 m ASL) and was connected with a glass fibre to the laboratory where the light source, spectrometer and other components were located. Three reflectors were positioned at different heights (896.5 m, 927 m and 959 m ASL) to the NNE of the telescope on a tower at the "Großer Felberg" Mountain (see Section 2.1) at a distance of

1.48 km from the light source. A fourth reflector was positioned on the "Großer Feldberg" mountainside at a distance of 1.23 km from the light source at 872m ASL. The overall measurement accuracies are 2%, 2% and 10% for $NO_2$, $O_3$ and $NO_3$, respectively, and are dominated by uncertainties in the absorption cross sections used. A schematic showing the light paths in relation to the site topography is shown in Fig. S2 of the supplementary information.

### 2.2.5 Particle properties

Particle size information was obtained using MoLa instruments (see above). A Fast Mobility Particle Sizer (FMPS 3091, TSI, Inc.), and an Aerodynamic Particle Sizer (APS 3321, TSI, Inc.) as well as an Optical Particle Counter (OPC 1.109, Grimm) covered a particle size range from 5.6 nm to 32 μm. The aerosol surface area (ASA) used for calculating rates of trace gas uptake was calculated from the combined FMPS and APS datasets or from the FMPS data alone (05.09 – 09.09). Generally, the bulk of the surface area (>75 %) was found in particles of diameter < 450 nm as measured by the FMPS.

### 25 2.2.6 Radiosoundings

The planetary boundary layer height was determined from radiosondes (GRAW, DFM-06), measuring temperature, relative humidity and position as already described for this campaign (Berkes et al., 2015). Each day, four to ten radiosondes were launched from the summit of the Kleiner Feldberg starting approximately one hour before sunrise and ending one hour after sunset. From these data, the potential temperature was calculated and used to determine the type of the boundary layer

(stable, neutral, turbulent).



## 3 Results and discussion

### 3.1 Meteorological conditions

During the 25 days of the PARADE campaign, the meteorological conditions were quite variable (see Fig. 1). As previously outlined (Phillips et al., 2012), the campaign can be separated broadly into 3 periods which are associated with different air

mass origins and the arrival of cold fronts. The first part of the campaign from the 15.08 to the 26.08 was characterized by a relatively high temperature (up to 25 °C) during the day and a variable day to day humidity and wind direction. The air originated from the sector between South and West, and back trajectory calculations using HYSPLIT (Draxler and Rolph, 2011) showed that it was located over the continent during the previous 48 hours. Exceptions to this pattern occurred on a few days when air masses passed over the English Channel. The marine influence could be traced through the presence of

enhanced $ClNO_2$ (measured using chemical ionisation mass spectrometry) during the night and early morning (Phillips et al., 2012).

The arrival of a cold front originating from the Atlantic on the evening of the 26.08 marks the beginning of the second period of the campaign. Immediately following the arrival of this front, the day time temperature decreased approximately by 10 °C and from the 26.08 to the 27.08, the relative humidity was close to 100 % and the mountain top was frequently in clouds.

The average daily temperature then slowly increased to reach 20 °C on the 03.09 in the afternoon. 48-hour back trajectories showed that from the 28.08 to the 01.09 the air was influenced by the Atlantic and the Benelux region. From the 01.09 to the 03.09, the 48-hour air mass origin was much closer to the site and stayed over the European continent. The arrival of another front on the evening of the 02.09 caused the temperature to drop again by 5 °C. Until the end of the campaign, the air originated from the UK and was characterised by low levels of solar radiation and high humidity as the mountain top was

frequently in clouds and fog.

### 3.2 $NO_2$, $NO_3$, and $N_2O_5$ mixing ratios

The $NO_3$, $N_2O_5$, $NO_2$ times series as well as that of $O_3$, NO and aerosol surface area (ASA) are shown in Fig. 2. The mixing ratios of all measured nitrogen species were highly variable due to the close proximity of anthropogenic activity and its spatial / temporal heterogeneity (Crowley et al., 2010; Phillips et al., 2012). $NO_2$ mixing ratios were well above the detection

limit during the whole campaign and ranged from 0.5 ppbv to more than 20 ppbv, with a campaign average of 2.7 ppbv during the day and 2.6 ppbv during the night. The highest, plume-like values of > 15 ppbv were spread over time periods of a few hours, occurred both day and night and were associated with polluted air masses arriving from the South-East (direction of Frankfurt). During passage of the cold fronts (vertical, blue lines in Fig. 2) the mountain-top was often in cloud with measured relative humidity close to 100 %. No $N_2O_5$ or $NO_3$ could be detected on these nights and $O_3$ levels were reduced to

background levels of ~25 ppbv during the following days. The low levels of solar radiation and photochemical activity reduced the maximum, daytime mixing concentrations of NO to ~ 0.5 ppbv.



As $NO_2$ is necessary for the formation of $NO_3$ and $N_2O_5$ (R1 and R4) and also because high levels are indicative of more polluted air masses which are usually associated with shorter $NO_3$ lifetimes, we examine the dependence of $NO_2$ on the local meteorological situation in more detail. In Fig. 3 we plot the relative count frequency of the $NO_2$ mixing ratios indexed against wind direction. The North-East sector was the least represented with a time coverage period of just 6 hours. The most frequently encountered wind direction was West ($\approx$ 200 hours) but it is also where low $NO_2$ mixing ratios (< 2 ppbv, in black) were most frequently encountered. The sector East to South-South-West was characterised by frequent high $NO_2$ mixing ratios. It is the only sector which was associated with $NO_2$ mixing ratios greater than 10 ppbv, which reflects emissions originating from the three local cities of Frankfurt, Mainz and Wiesbaden. Air masses coming from this sector are expected to have the highest aerosol loading and levels of anthropogenic hydrocarbons.

$[NO_3]$ and $[N_2O_5]$ were highly variable during a single night and also from one night to the next, ranging from below the detection limit to 250 pptv for $NO_3$ and 3 ppbv for $N_2O_5$, which is significantly greater than previously measured at this site (Crowley et al., 2010). Precipitation periods were associated with $[NO_3]$ and $[N_2O_5]$ below the detection limit, which is a result of $N_2O_5$ uptake to droplets. The high variability in both $[NO_3]$ and $[N_2O_5]$ is partly due to the variability in their production rate. Fig. 4 (upper panel) plots the mixing ratio of $NO_3$ (measured between 19:00 and 06:00 and at relative humidity lower than 97 %) and its production rate ($P(NO_3)$, in ppt s$^{-1}$, lower panel) as calculated using Eq (1) against the local wind direction. Both production rate and mixing ratio of $NO_3$ show higher values for air masses coming from the Frankfurt and the Mainz/Wiesbaden sectors.

### 3.3    $NO_3$ and $N_2O_5$ lifetimes

The "steady state" method for estimating the lifetime of $NO_3$ is based on the assumption that, after a certain time following sunset, the $NO_3$ and $N_2O_5$ concentrations in a single air-mass build up to a quasi-constant value. Steady state is reached when the sum of direct and indirect loss rate constants of $NO_3$, $L(NO_3)$ in s$^{-1}$, balances its production rate, $P(NO_3)$ so that:

$$[NO_3]_{SS} = \frac{P(NO_3)+T_{in}(NO_3)}{L(NO_3)+T_{out}(NO_3)} \qquad \text{(Eq2)}$$

where $P(NO_3)$ and $L(NO_3)$ are terms for $NO_3$ chemical formation and loss, respectively and $T_{in}(NO_3)$ and $T_{out}(NO_3)$ represent the influence of transport of $NO_3$. Generally, the terms $T_{in}(NO_3)$ and $T_{out}(NO_3)$ are regarded as insignificant for radical species with chemical lifetimes on the order of minutes rather than hours or days. The simpler form, Eq (3), for the steady-state approximation has therefore been used frequently in analysing $NO_3$ measurements (Noxon et al., 1980; Platt et al., 1980; Allan et al., 1999) and has been examined in detail (Brown et al., 2003). At steady state, the $NO_3$ lifetime ($\tau_{ss}$) is then:

$$\tau_{ss}(NO_3) = \frac{[NO_3]}{k_1[NO_2][O_3]} = \frac{1}{L_{ss}(NO_3)} \qquad \text{(Eq3)}$$



The term $L_{ss}(NO_3)$ is the loss term of $NO_3$ corresponding to the steady state lifetime. The loss term $L(NO_3)$ is the sum of processes removing $NO_3$ and is often simplified as in Eq (4):

$$L(NO_3) \approx (k_6[NO] + \sum([VOC]_i k_i) + 0.25\, K_{eq}[NO_2]\, \bar{c}\, ASA\, \gamma_{N2O5})^{-1} \qquad \text{(Eq4)}$$

where $k_6$ is the rate constant for R6 ($2.6 \times 10^{-11}$ cm$^3$ molecule$^{-1}$ s$^{-1}$ at 298 K (Atkinson et al., 2004) and $K_{eq}$ is the equilibrium constant describing the relative concentrations of $NO_3$ and $N_2O_5$ under equilibrium for any given $NO_2$ concentration and temperature, and is given by $K_{eq} = k_4 / k_{-4}$. $\bar{c}$ is the mean, thermal velocity of $N_2O_5$, $k_i$ is the rate constant for reaction of $NO_3$ with a VOC, and $\gamma_{N2O5}$ is the uptake coefficient for $N_2O_5$ to aerosol with particle surface area ASA (cm$^2$ cm$^{-3}$). Expressions (3) and (4) ignore loss of $NO_3$ via heterogeneous uptake and loss of $N_2O_5$ via gas phase reactions, both of which are generally considered negligible, and considers $NO_3$ production path through oxidation of $NO_2$ by $O_3$ only. Later, we shall discuss the potential impact of other $NO_3$ production pathways.

Implicit to this analysis is the assumption that steady state is achieved when the air mass reaches the measurement site. The strongest, local sources of $NO_2$ and $O_3$ are likely to be the urban centres of Frankfurt, Mainz and Wiesbaden, all ~20 km distant in the southern sectors. As outlined above, there are no close, strong, continuous sources of anthropogenic pollution in the northern sectors. The $NO_3$ production rate as a function of wind direction in Fig. 4 confirms that the whole northern sector is relatively low in $NO_3$ precursors.

With typical wind speeds of between 2 and 5 m.s$^{-1}$ the transport times are at least 1 hour from the major pollution sources. The time required to achieve steady state depends on the $NO_3$ production term and the overall loss frequency and is shorter when $P(NO_3)$ is low and the removal rate high. Numerical simulations were performed to test the validity of the steady state approximation (Brown et al., 2003) and details are given in the supplementary information (Fig. S3). The results indicate that achievement of steady-state, can indeed take longer than 1 hour, when $NO_3$ lifetimes are long, especially when considering air masses with high $NO_2$ and $O_3$, e.g. as occasionally encountered from the southern sectors. During the first hours of nightime, use of equation 3 may thus result in *underestimation* of the steady-state lifetime of $NO_3$.

The $NO_3$ lifetime can also be calculated from the production rate of $NO_3$ and the time derivatives of $[NO_3]$ and $[N_2O_5]$ as in Eq (5) and thus does not require steady state to have been acquired (McLaren et al., 2010). The lifetime thus calculated, $\tau_{DER}$ $(NO_3)$ is given by:

$$\tau_{DER}(NO_3) = \frac{[NO_3]}{k_1[NO_2][O_3] - \frac{d[NO_3]}{dt} - \frac{d[N_2O_5]}{dt}} \qquad \text{(Eq5)}$$

As in the steady state approximation, this method assumes that only chemical losses impact on $NO_3$ and $N_2O_5$ mixing ratios (negligible transport in and out of the sampled air mass) and that the sink terms are constant over the time step considered.

The campaign time series of the steady state $NO_3$ lifetime is presented in Fig. 5 (top panel). The night to night variability of $\tau_{ss}(NO_3)$ is very high throughout the 3 weeks of the campaign, with average values of 200 s ($\pm$ 100 s) for $NO_3$. The longest lifetimes (~ 1 hour for $NO_3$) were observed during 3 consecutive nights on the 30.08, 31.08 and 01.09.



In Fig. S4 of the supplementary information, we plot the $NO_3$ steady state lifetime, $\tau_{ss}(NO_3)$ and the lifetime calculated using the derivatives method, $\tau_{DER}(NO_3)$. Both calculations agree well for the lowest $NO_3$ lifetimes (first period of the campaign from the 15.08 to the 26.08). This is expected, as the steady state condition is readily fulfilled when the losses of $NO_3$ (or $N_2O_5$) are rapid. During the nights 30.08, 31.08 and 01.09, $\tau_{ss}(NO_3)$ is up to two times greater than $\tau_{DER}(NO_3)$. This is clearly not related to the effects outlined above which apply to the first hours of darkness, but as we discuss later, is associated with deviation from steady state during these periods.

The derivatives method does not provide data for the whole campaign. $\tau_{DER}(NO_3)$ are sometime very scattered and sometimes negative which arises from noise on the derivative terms. Because the error made by assuming steady state is generally small, the analysis and discussion below is based on the lifetimes obtained by the steady-state analysis.

The wind-rose in the bottom panel of Fig. 5 indicates that high values of $\tau_{ss}(NO_3)$ ($\approx$ 1 h) were encountered irrespective of wind direction. This is initially surprising, as long $NO_3$ lifetimes values are not expected in air masses originating from the highly polluted sectors.

The short term (hour to hour) variability was very high as already described in (Crowley et al. 2010) for this site. They describe the evolution of $\tau_{ss}(NO_3)$ over a single night in which the lifetime increased slowly after sunset to reach a roughly constant value around midnight. As steady state was predicted to be achieved on a much shorter time scale, and because other loss processes involving NO or heterogeneous loss of $N_2O_5$ were too slow, the increase in $[NO_3]$ and $\tau_{ss}(NO_3)$ was interpreted as being due to a slow decrease in the concentration of VOCs, though measurements of VOCs were not available to confirm this. During the PARADE campaign a number of $VOC_S$ and the aerosol surface area were measured, allowing us to calculate the overall loss rate constant of $NO_3$ using Eq (4). As only a subset of the suite of organic species likely to be present at the site was measured, this calculation will provide a lower limit for the overall $NO_3$ loss rate (and thus an upper limit for its real lifetime).

### 3.3.1 Loss of $NO_3$ via reaction with NO

With an average $O_3$ mixing ratio of about 40 ppbv, the NO lifetime at nighttime is about 5 min, so that, in the absence of local sources and photochemical degradation of $NO_2$, the NO mixing ratio will be close to zero within an hour of sunset. This was the case for most of the nights during PARADE and NO measurements were below the (5 pptv) detection limit. On some nights however, as exemplified in Fig. 6, 10 – 20 pptv of NO were measured and between 21:00 and sunrise, $\tau_{ss}(NO_3)$ (upper panel, black line) was consistent with that calculated from the NO reactivity as $k_6[NO]$ (upper panel, red line). During the period from 19:00 to 21:00 the lifetime of $NO_3$ attributable to reaction with NO, $\tau(NO)$, is greater than $\tau_{ss}(NO_3)$, probably because steady state is not yet reached for $NO_3$. A 2 hour period necessary to reach steady state is consistent with the results obtained from the steady state simulation described in **Sect. 3.3**. Fig. 6 shows that the non-zero NO mixing ratios encountered on this night were in air masses associated with low wind speeds, and may be a result of vehicle use on local roads or soil emissions.



### 3.3.2    Heterogeneous removal rate

The overall loss rate constant for $NO_3$ contains a contribution from indirect loss via the heterogeneous removal of $N_2O_5$ to particles as described by Eq (4). In Fig. 7, we plot the calculated loss rate constant ($s^{-1}$) for $NO_3$ resulting from the uptake of $N_2O_5$ to particles ($L_{HET}$ in blue) and compare it to the overall loss rate constant in steady state ($L_{ss}$, open circles) for two

different nights. A simple, temperature dependent value of the uptake coefficient ($\gamma = 0.244 - 7.9 \times 10^{-4}\ T$) was used as recommended by IUPAC. This expression is based on laboratory studies and results in a value at 283 K of ~$2 \times 10^{-2}$.

During periods when the $NO_3$ lifetime is long (e.g. $\tau_{ss}(NO_3) > 2000$ s or $L_{ss}(NO_3) < 5 \times 10^{-4}\ s^{-1}$) as on the second period of the night of the 30.08 (lower panel), the losses of $N_2O_5$ to particles can account entirely for the observed lifetimes, or even exceed the steady state reactivity. On this night, NO was below the instrumental detection limit and was not considered for

$NO_3$ losses. During periods when the $NO_3$ lifetime is shorter (e.g. on the 05.09, upper panel) the indirect, heterogeneous loss represents only a small fraction of the overall loss. More sophisticated parameterisations of the uptake coefficient including e.g. the nitrate and chloride content of the particles and also organic coatings exist (Bertram and Thornton, 2009), often fail to reproduce in situ measurements (Bertram et al., 2009) and indicate that $\gamma$ is a large source of uncertainty in these calculations. However, even the use of values as large as 0.1 does not reproduce the shorter lifetimes and, as previously

discussed for this site (Crowley et al., 2010), the remaining losses are likely due to direct reactions of $NO_3$ with VOCs.

### 3.3.3    Loss of $NO_3$ via reaction with VOCs

The VOCs measured during PARADE are listed in Table 2 together with their rate constants (at 298 K) for reaction with $NO_3$. Among the trace gases listed, only a few of them (at high concentrations or with large rate constants for reaction with $NO_3$) contributed significantly to the loss of $NO_3$ and are highlighted in bold. Further discussion will focus on these VOCs.

Apart from isoprene (up to 200 pptv), the mixing ratios of VOCs which are reactive towards $NO_3$ were generally below 100 pptv (see supplementary information Fig. S5). A clear diel cycle was observed for isoprene and during the warmer first and last parts of the campaign daytime mixing ratios were up to 5 times higher than during the night. For the monoterpenes the diel cycle was less pronounced and on two nights (04.09-05.09 and 05.09-06.09) large mixing ratios were observed during the night (e.g. > 50 pptv for myrcene and limonene), which were not accompanied by increases in isoprene. This may be the

result of emission of terpenes into an especially stable, shallow boundary layer. As expected, during colder, rainy periods, the BVOCs were less abundant and showed a weaker day/night cycle.

For each monoterpene, the $NO_3$ loss rate constant for reaction with the measured VOCs ($L_i = k_i[VOC]_i$) where $k_i$ is the rate constant for each VOC) is plotted together with the steady state $NO_3$ total reactivity ($L_{ss}(NO_3)$) and the VOCs summed reactivities (including alkanes and alkenes) in Fig. 7 and Fig. S6. The rate coefficients for reaction of $NO_3$ with the VOCs

were derived from the temperature dependent expressions given by IUPAC and were calculated at each time step of the dataset. Fig. S6 shows that, over almost the whole campaign, the steady state loss rate constants $L_{ss}$ (open circles) are significantly lower than those calculated by summing the individual contributions of the VOCs ($L_{VOCs}$, dashed line). Indeed,



during the 3 nights in which very long $NO_3$ lifetimes were encountered, $L_{ss}$ is approximately 4 to 5 times lower than that expected from the summed VOCs. In other words, within the steady-state framework used here and using the $NO_3$ production term, $k_1[NO_2][O_3]$, the observed mixing ratios and lifetimes of $NO_3$ are incompatible with the VOC measurements.

The divergence between the steady-state loss rates and that obtained by summing the losses with each VOC is exacerbated when one considers that the VOCs measured in PARADE cover only a fraction of those present in the air so that $L_{VOC}$ must strictly be regarded as a lower limit. As exemplified in Fig. 7 and Fig. S6, summing the individual loss rate constants due to reaction of $NO_3$ with NO ($L_{NO}$), the complete set of VOCs ($L_{VOC}$) and due to indirect loss via heterogeneous uptake of $N_2O_5$ to available particles ($L_{HET}$), results in an air-mass reactivity that is larger than that derived from the steady state calculations.

i.e. the $NO_3$ lifetime is longer than expected (or its concentration higher). Generally, the opposite is observed and missing reactivity is assigned e.g. to VOCs that were not measured. Below, we examine possible explanations for this, including the breakdown of the steady state assumption, the potential role of other chemical routes to $NO_3$ generation and meteorological effects.

### 3.3.4    $NO_3$ Production rate

So far, our calculations have been based on the assumption that $NO_3$ is generated solely in the reaction between $NO_2$ and $O_3$ (R1, see Sect. 3.3). If $NO_3$ is only formed via R1, the uncertainty associated with $P(NO_3)$, propagated from those of $[NO_2]$ and $[O_3]$ is approximately 8 %. The good agreement between several devices for measurement of both $NO_2$ and $O_3$ indicates that the term $k_1[NO_2][O_3]$ is indeed well defined. As indicated in Sect. 1, other sources of $NO_3$ are known, but are generally regarded as negligible. The reaction of OH with $HNO_3$ (R2) generates $NO_3$ at ~100 % yield but, even assuming nighttime

concentrations of hydroxyl radicals of $1 \times 10^5$ molecule $cm^{-3}$ and 2 ppbv of $HNO_3$, the $NO_3$ production rate is only ~ 1 % of that from R1.  In the absence of sunlight, the formation of $NO_3$ from the photolysis (R3) of halogen-nitrates $XONO_2$ (where X = Cl, Br or I) will not contribute at nighttime. A further, possible source of nighttime $NO_3$ and cause for error in steady-state lifetime estimations could be the rapid re-cycling of $NO_3$ from the products formed in its initial reaction with BVOCs, as has been seen for the OH radical (Lelieveld et al., 2008; Taraborrelli et al., 2012). The organic nitrates formed as first

generation products when $NO_3$ reacts with biogenic VOCs are however believed to be chemically stable on these time scales and there is no obvious pathway for rapid re-release of $NO_3$ once formed.

We now consider a further, nighttime process forming $NO_3$ via the oxidation of $NO_2$ by stabilized Criegee intermediates (sCI), which are formed in the atmosphere by the reaction of $O_3$ with biogenic and anthropogenic alkenes (Johnson and Marston, 2008). This may be especially relevant when observing apparently long $NO_3$ lifetimes (i.e. high concentrations) in

the presence of reactive VOCs.

The possibility of $NO_3$ generation via sCI reaction with $NO_2$ was first raised several years ago (Fenske et al., 2000; Presto and Donahue, 2004) though in the early studies, which were unable to detect Criegee radicals directly, uncertainty regarding the products and the rate constant were large. In recent work (Welz et al., 2012) on the reaction between the simplest sCI





($CH_2O_2$) and $NO_2$ a large rate constant ($7^{+3}_{-2}$ $10^{-12}$ $cm^3$ molecule$^{-1}$ s$^{-1}$) was determined. The authors hypothesised that the reaction between other sCI and $NO_2$ might have similar rate constants. A further study (Stone et al., 2014) reports a rate constant ($1.5 \pm 0.5 \times 10^{-12}$ $cm^3$ molecule$^{-1}$ s$^{-1}$) for the same reaction and detected HCHO as product, consistent with an O-atom transfer from sCI to $NO_2$ to form $NO_3$. Subsequently, the formation of $NO_3$ (and $N_2O_5$) resulting from the photolysis at

248 nm of a mixture of $CH_2I_2$, $O_2$ and $NO_2$ was attributed to the reaction between $CH_2O_2$ and $NO_2$ (Ouyang et al., 2013). Whether sCI can contribute significantly to nighttime $NO_3$ formation or not depends on the assumption that all sCI react with a similar rate coefficient to $CH_2O_2$, that $NO_3$ is formed at high yield, and that sufficient sCI are present in the nighttime boundary layer. In Table 3, we assess the three reactions that could have contributed to $NO_3$ formation during PARADE. We take a generic rate coefficient for sCI + $NO_2$ that lies between the literature determinations (Welz et al., 2012; Stone et al.,

2014) and assume 100 % product yield for formation of $NO_3$.

Table 3 shows that, whereas the reaction between OH and $HNO_3$ can safely be neglected, the reaction between sCI and $NO_2$ can represent a significant fraction of the total rate of production of $NO_3$ if sCI are present at ~ 0.1 pptv during night. This reflects the fact that, although Criegee radicals are expected to be present at concentrations 4 – 6 orders of magnitude less than those of $O_3$, the rate constant for reaction with $NO_2$ is ~ 5 orders of magnitude larger.

The rate of formation of sCI in the boundary layer depends both on the concentration and nature of the organics reacting with $O_3$ to form it and is expected to be very variable. Likewise, the sink reactions of sCI are difficult to predict, though reactions with water vapour, $SO_2$ and $NO_2$ are expected to be important (Vereecken et al., 2012). Boundary layer mixing ratios of sCI are thus associated with great uncertainty. Based on a steady state approach, sCI mixing ratio up to 0.03 pptv have ben calculated for the PARADE campaign (Bonn et al., 2014), i.e. within the range of values used in Table 3. Recent

measurements in a Boreal forest environment estimated sCI mixing ratios to be 0.0025 to 0.04 pptv (Taipale et al., 2014) and sCI have been tentatively identified (Mao et al., 2012) as the source of an interfering signal in ambient measurements of OH that may approach 0.2 pptv (Novelli et al., 2014). As not all ambient sCI will decompose to OH within such instruments, the wider implication is that pptv amounts of sCI may be present.

To calculate the influence of sCI induced formation of $NO_3$ during the PARADE campaign we recalculated the $NO_3$

production rate assuming two different sCI concentrations at the limits of the range listed in Table 3 and using the rate constant given. In Fig. 8 we display the calculated steady-state lifetime without sCI reactions forming $NO_3$ (black symbols) and the effect of using a low sCI concentration ($\tau_{LsCI}$, blue symbols) or a high concentration ($\tau_{HsCI}$, red symbols). The resultant reduction in the calculated steady-state lifetime of $NO_3$ is significant when using the higher values of [sCI]. For example, on the 31.08, the largest value of $\tau_{ss}(NO_3)$ would be reduced from 3000 s (black data points) to 2000 s (red data

points) when assuming 0.1 pptv of sCI. However, in order to bring the steady-state lifetime into agreement with that based on the VOC measurements (green symbols), a sCI mixing ratio of 4 pptv would be required. Our present knowledge of Criegee formation and loss mechanisms under atmospheric conditions precludes accurate assessment of their role in forming $NO_3$ at nighttime, though sCI mixing ratios as large as 4 pptv appear (at present) unlikely, though not impossible.





### 3.3.5 Meteorological considerations

We have shown that, while uncertainties in some parameters required for calculation of $NO_3$ lifetimes exist, they are likely to be insufficient to explain the high concentrations (and lifetimes) of $NO_3$ during the nights 30.08 to 01.09. Close examination of the $NO_3$ lifetime on these nights reveals an especially large hour-to-hour variability with $\tau_{ss}(NO_3)$ values increasing by e.g.

500 to 600 % over the course of an hour. Simultaneously, $\tau_{ss}(N_2O_5)$ values (not shown) increase by 200 %. This smaller increase in $\tau_{ss}(N_2O_5)$ is due to a simultaneous decrease of $[NO_2]$ by 50 % thus shifting the equilibrium towards $NO_3$. Such short time variability cannot be attributed to a decrease in the mixing ratios of reactive trace gases, which were much less variable (see above). We therefore consider the possibility that the long lifetimes observed on these nights are related to sampling air masses from a low-lying residual layer which, especially at low wind speeds, is decoupled from ground

emissions of NOx and VOCs, allowing $NO_3$ and $N_2O_5$ to build up to higher levels. This would also explain the lack of dependence on wind direction and the high temporal variability.

The hypothesis that sampling of residual-layer air is responsible for the long steady-state lifetimes of $NO_3$, and their apparent incompability with the VOC measurements is examined in detail in Fig. 9. Here, the time profiles of $\tau_{ss}(NO_3)$, $NO_2$, $O_3$, $CO_2$, and ASA and some meteorological parameters are plotted for the night of the 30.08. The night from the 30.08 to the 31.08

can be divided into two distinct periods. Between 21:00 and 00:30 (1$^{st}$ period) $\tau_{ss}(NO_3)$ was roughly constant at an average value of about 150 s. After this, we observed a large increase in the $NO_3$ steady-state lifetime between ~00:30 and 01:30, followed by period 2 (01:30 until dawn) in which $\tau_{ss}(NO_3)$ was consistently close to 3000 s.

The transition from the shorter (period 1) to longer lifetimes (period 2) is accompanied by a decrease in the $NO_2$ mixing ratio from about 2.5 to 1 ppbv, a ~2 ppm decrease in $CO_2$, an increase in $O_3$ from ~15 to 25 ppbv and a ~ 30 % increase in particle

surface area. Changes in the local wind direction (at ~ 20 m height) between the clean Northern sector to the anthropogenically polluted South sector during this 2$^{nd}$ period did not impact the NOx levels or the $NO_3$ lifetime.

These observations are consistent with air from the residual layer being sampled during the 2$^{nd}$ period. Higher levels of $O_3$ and particles, formed during the previous day in the turbulently mixed boundary layer, are expected in the residual layer compared to the shallow, nocturnal boundary layer in which dry deposition is important for both. Conversely, levels of $NO_2$,

formed from oxidation of ground level NO emissions are expected to be higher in the lower levels (Stutz et al., 2004; Brown et al., 2007) and thus decrease when sampling air from the residual layer. The second period is also marked by a significant drop in relative humidity (from ~ 80 to 65 %) and an increase in the temperature of ≈ 1 °C. The former is expected as relative humidity decreases above the residual layer. A correlation between increasing $NO_3$ lifetimes and temperature has been observed previously (Crowley et al. 2010) and was attributed to downward movement of higher air masses during an

inversion.

Fig. 10 (upper panel) shows profiles of the potential temperature and relative humidity within the first period at 19:00 and within the second period at 04:00 on 30.08 to 31.08. The temporal development of the temperature and the relative humidity profiles agrees well with the near-surface observations. The profiles of the potential temperature show a change from stable



stratification (potential temperature increase with height) to neutral stratification (potential temperature is constant with height) which is an additional indication that the measurements within period 2 were obtained within the residual layer. The same temporal development of the potential temperature profiles was observed 24 hours later (upper panel), when again the $NO_3$ lifetime was enhanced. On both nights, the lack of a switch in sign of the RH gradient with height at 04:00 contrasts with the profiles at 19:00 and 06:00 which show clear evidence for a switch from boundary-layer to residual layer some 50 to 100 m above the summit. Between the 31.08 and 02.09 we observed no nocturnal build-up of $CO_2$, which strongly contrasts the periods prior to and after this period (see Fig. S7) and again indicates that this period with the longest $NO_3$ lifetimes is associated with residual layer air.

So far, we have assumed that $NO_3$ (or $N_2O_5$) levels aloft (i.e. in the residual layer) are enhanced w.r.t. the boundary layer. As $NO_3$ (and also $NO_2$ and $O_3$) was measured by long path absorption spectroscopy (LP-DOAS) at different heights, this dataset should give some insight into the occurrence and extent of vertical gradients in $NO_3$ concentrations. In Fig. 9 we plot a data set obtained using the LP-DOAS set-up over the same period. In the first period, the DOAS measurements of $NO_3$ at the highest altitude were significantly larger (factor of ~5 at 24:00) than those measured by the CRD. This difference in concentration disappeared during the second phase when both instruments measured high $NO_3$, thus supporting the concept that the top of the mountain is in the residual layer during this period. Below, we compare the CRD and DOAS measurements in more detail.

### 3.3.6 Comparison of $NO_3$ lifetimes derived from CRDS and LP-DOAS instruments

As mentioned in Sect. 2.2 the LP-DOAS measurements were conducted using a total of 4 light paths. For this analysis we compare data retrieved from only the lowest (835 to 872 m ASL) and highest (835 to 959 m ASL) levels (see Fig. S2). By comparison, the height of the CRDS inlet was ~ 838 m ASL. We refer to the lower and higher levels as DOAS,1 and DOAS,4 respectively. Steady-state $NO_3$ lifetimes were calculated in the same way as described above for the CRDS dataset, using the $NO_2$ and $O_3$ data obtained from the DOAS instrument at those two different levels. Because of the short horizontal (1.5 km) and vertical (~125 m) range of the DOAS light paths, the integrated temperature over both light paths is assumed to be the same as the temperature measured at the Kleiner Feldberg hill top and these values were used to calculate the rate constant ($k_1$) for $NO_3$ production. Note that differences in temperature of a few Kelvin do not impact significantly on the rate constant, $k_1$.

In Fig. 11 we plot the $NO_3$ mixing ratios (upper panels) and steady state lifetimes (lower panels) obtained on two different nights, 06.09 and 02.09, by the CRDS and LP-DOAS instruments. While the night of the 06.09 shows large differences between the mixing ratios reported by the CRDS and DOAS instruments, and a strong vertical gradient in $NO_3$, good agreement (i.e. a weak or no gradient) is observed on the 02.09. The DOAS measurements of the $NO_2$ (Thieser et al., 2015) and $O_3$ mixing ratios (see supplementary information Fig. S8) revealed little or no difference between the highest and lowest light paths over the whole campaign, so that the $NO_3$ production term was roughly constant with altitude. The clear dependence of the $NO_3$ mixing ratio on height on the 06.09 is thus due to variation in the $NO_3$ loss term. The lower, left



panel of Fig. 11 indicates a factor of up to ~5 increase in $NO_3$ lifetime when comparing the LP-DOAS data at the highest level to the CRDS measurements.

In Fig. 12 (left panel) we plot the campaign $NO_3$ mixing ratios measured by the CRDS and DOAS,4 versus the DOAS,1 dataset. All data which were above the detection limit (10 ppt for LP-DOAS and 5 ppt for the CRDS) were included. The coloured lines (black, blue and red) represent slopes of 0.5, 1 and 2 respectively. Nearly all of the black data points $[NO_3]_{DOAS,4}$ have slopes between 1 and 2, indicating that, on average $[NO_3]$ is higher aloft, consistent with previous observations of positive vertical gradients in $NO_3$ (Stutz et al., 2004; Brown et al., 2007). At large mixing ratios, i.e. for $[NO_3]_{DOAS,1} > 100$ pptv, the altitude dependence on the $[NO_3]$ is weaker and the black points are closer to the blue line (slope of 1). The ratio between $[NO_3]_{CRDS}$ and $[NO_3]_{DOAS,1}$ also shows some dependence on the $NO_3$ mixing ratio. When the $NO_3$ mixing ratios are low, the ratio CRDS / LP-DOAS is also low and generally less than 0.5. In extreme cases, the $[NO_3]_{CRDS}$ values are close to a few pptv or below the detection limit while $[NO_3]_{DOAS,1}$ measured as high as 40 to 50 ppt.

Both $\tau_{ss}(NO_3)_{CRDS}$ and $\tau_{ss}(NO_3)_{DOAS,4}$ datasets (Fig. 12, right panels) show the same trends as the one described for $[NO_3]$. At low $\tau_{ss}(NO_3)_{DOAS,1}$ the black and red points are respectively closer to the black and red lines and as $\tau_{ss}(NO_3)_{DOAS,1}$ increases to the highest values, the data points are all closer to the blue line. The apparent dependence of the $NO_3$ mixing ratios and lifetime on the $NO_3$ concentrations reflects the fact that, when $NO_3$ is high both instruments are sampling the residual layer in which there are no significant gradients in $NO_3$.

In summary, the data shown in Fig. 9 and the LP-DOAS measurements of $NO_3$ at different altitudes provide compelling evidence for a low-lying residual layer being responsible for the occasional observation of high $NO_3$ mixing ratios seen by the ground level CRDS instrument and the associated long steady-state lifetimes. This layer will be most stable and thus best decoupled from the underlying boundary layer when wind-speeds are low.

When sampling from a low-lying residual layer, we would expect that the observed increase in $O_3$ and ASA and simultaneous decrease in $NO_2$ and RH would be accompanied by a decrease in those biogenic trace gases, which are reactive towards $NO_3$. Indeed, if we take the measured $NO_3$ mixing ratio of 200 pptv on the 02.09, we calculate that the lifetime of biogenics such as e.g. limonene should be of the order of only a few minutes in the residual layer. The simultaneous measurement of both high $NO_3$ steady-state lifetimes and terpene mixing ratios can now be explained if we consider that the terpene emissions are only local, i.e. from trees that are close to the top of the mountain (and our inlet) and thus also within the residual layer. Such local emission (e.g. within 30 m of the inlet) does not allow the $NO_3$ + terpene chemistry to go to steady-state so that only a fraction of the $NO_3$ (and terpene) are consumed. The extent of reaction will be related to the location of the emission relative to the inlet and local wind-speeds and is likely to be highly variable.

## 4 Conclusions

We observed great variability in $NO_3$ concentrations and lifetime at a mountain site in South-West Germany. Measurements of $NO_3$, its precursors and its sink reactions (both direct and indirect) enabled assessment of the processes (both chemical



and meteorological) influencing its steady-state lifetime. We found that, during several nights, the observed, steady-state lifetime was frequently larger than expected based on measured gas-phase reactants such as VOCs and NO. We have shown that an enhancement in the $NO_3$ production term via the reaction between Criegees and $NO_2$ may be significant, but unlikely to explain the discrepancy. The periods with the highest apparent lifetimes are associated with the presence of a low-lying

5    residual layer which encompasses the mountain top and results in the $NO_3$ (and $N_2O_5$) mixing ratios and lifetimes larger than those calculated from measured sources and sink terms.





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





**Acknowledgements**

This work was carried out in part fulfilment of the PhD of NS at the Johannes-Gutenberg-University, Mainz. We thank the following: Simone Stöppler & Thomas Elsinger of 4 the "Hessischer Rundfunk" for mounting the DOAS retro-reflectors on

5   the tower at the 5 Großer Feldberg. Dupont for proving a sample of the FEP   suspension used to coat the inlets and cavities. Pablo Hidalgo for generation of Land Use maps derived from the CORINE Land Cover database. The HLUG for use of meteorological measurements and Heinz Bingemer for logistical support throughout the planning and execution of the campaign.



**Table 1. Instruments deployed during PARADE.**

| Species | Instrument | Inlet position |
|---|---|---|
| $NO_3$ and $N_2O_5$ | CRDS | Tower |
| $NO_3$ | LP-DOAS | Roof* |
| $NO_2$ | CRDS | Tower |
| NO | CLD | Tower |
| VOCs | GC-MS/GC-FID | Tower |
| $O_3$ | AirPointer | MoLa |
| Aerosol Surface Area | FMPS/APS | MoLa |

5   **Notes:** *The telescope of the LP-DOAS setup were located on the roof, about 2m below the inlets of the point

measurements. The retroreflectors were at different heights at a distance of ~1.5 km towards the North (see Sect. 2.2.4).



**Table 2. VOCs measured using GC-MS and GC-FID during PARADE**

| GC-MS VOCs | $k(NO_3 + VOC)^a$ | Maximum mixing ratio (pptv) | GC-FID VOCs | $k(NO_3 + VOC)^a$ | Maximum mixing ratio (ppbv) |
|---|---|---|---|---|---|
| **isoprene** | $\mathbf{7.0 \times 10^{-13}}$ | 220 | Ethane | $<1 \times 10^{-17}$ | 7.2 |
| **α-pinene** | $\mathbf{6.2 \times 10^{-12}}$ | 170 | Ethene | $2.1 \times 10^{-16}$ | 15 |
| **myrcene** | $\mathbf{1.1 \times 10^{-11}}$ | 70 | Propane | $<7 \times 10^{-17}$ | 2.5 |
| **limonene** | $\mathbf{1.22 \times 10^{-11}}$ | 130 | Propene | $9.5 \times 10^{-15}$ | 20 |
| p-cymene | $1.0 \times 10^{-15\ b}$ | 20 | $i$-Butane | $1.1 \times 10^{-16\ b}$ | 1.3 |
| benzene | $<3 \times 10^{-17\ b}$ | 230 | $n$-Butane | $4.6 \times 10^{-17}$ | 1.4 |
| toluene | $7.0 \times 10^{-17\ b}$ | 420 | cis-2-Butene | $3.5 \times 10^{-13}$ | 0.6 |
| ethylbenzene | $<6 \times 10^{-16\ b}$ | 80 | $i$-Pentane | $1.62 \times 10^{-16\ b}$ | 1 |
| m-xylene | $2.6 \times 10^{-16\ b}$ | 130 | $n$-Pentane | $8.7 \times 10^{-17\ b}$ | 1 |
| p-xylene | $5.0 \times 10^{-16\ b}$ | 65 | 1-Pentene | $1.5 \times 10^{-14\ b}$ | 1.9 |
| o-xylene | $4.1 \times 10^{-16\ b}$ | 75 | Butadiene | $1.0 \times 10^{-13\ b}$ | 10.6 |

**Notes:** [a]Rate constants ($cm^3$ molecule$^{-1}$ s$^{-1}$) at 298 K taken from the latest evaluation (Atkinson et al., 2006; IUPAC, 2015) unless noted otherwise. [b]Rate constants ($cm^3$ molecule$^{-1}$ s$^{-1}$) taken from (Atkinson and Arey, 2003).



**Table 3. Reactions producing $NO_3$.**

| Reaction | Mixing ratio 1st reactant | Mixing ratio 2nd reactant | Rate coefficient (298K)[a] | P($NO_3$)[b] |
|---|---|---|---|---|
| $O_3 + NO_2$ | 20 – 50 ppbv | 0.5 – 5 ppbv | $3.5 \times 10^{-17}$ | $(0.9 – 220) \times 10^{-3}$ |
| $OH + HNO_3$ | $(1 – 10) \times 10^{-2}$ pptv | 0.5 – 5 ppbv | $1.5 \times 10^{-13}$ | $(1.9 – 190) \times 10^{-5}$ |
| $sCI + NO_2$ | 0.01 – 0.1 pptv | 0.5 – 5 ppbv | $5.0 \times 10^{-12}$ | $(0.6 – 6) \times 10^{-3}$ |

**Notes:** [a]Units of $cm^3$ molecule$^{-1}$ s$^{-1}$. [b]Units of pptv s$^{-1}$





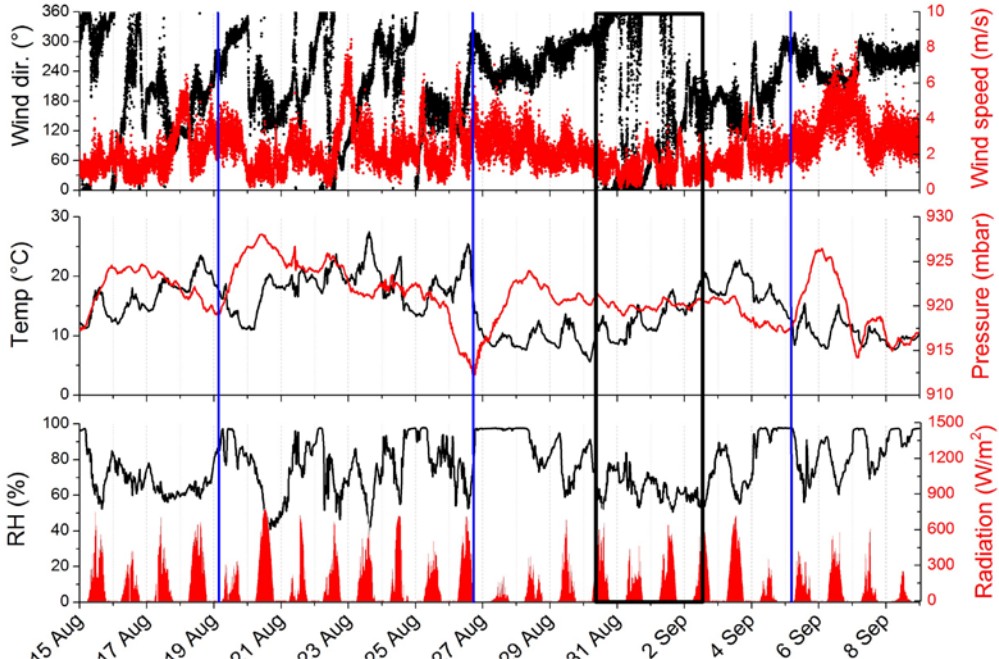

**Fig. 1.** Meteorological parameters measured during the PARADE campaign. The vertical, blues lines represent the arrival of cold fronts. The black box corresponds to the time period in which the highest $NO_3$ steady state lifetimes were measured.



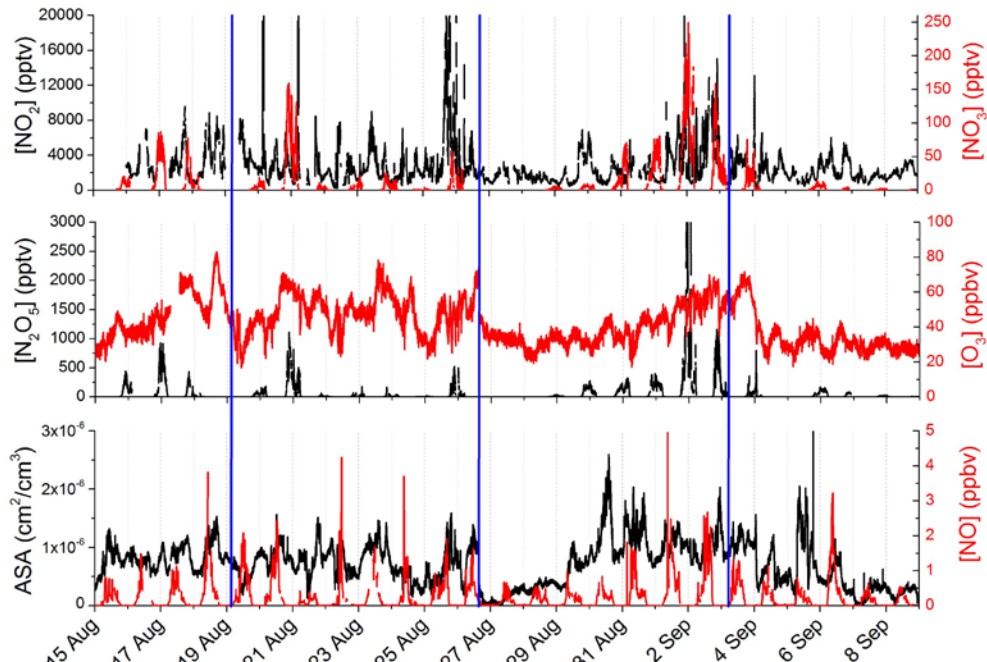

**Fig. 2.** NO$_2$, NO$_3$, N$_2$O$_5$, O$_3$ and NO volume mixing ratios and aerosol surface area (ASA) measured during the PARADE campaign. The vertical, blue lines represent the arrival of cold-fronts.





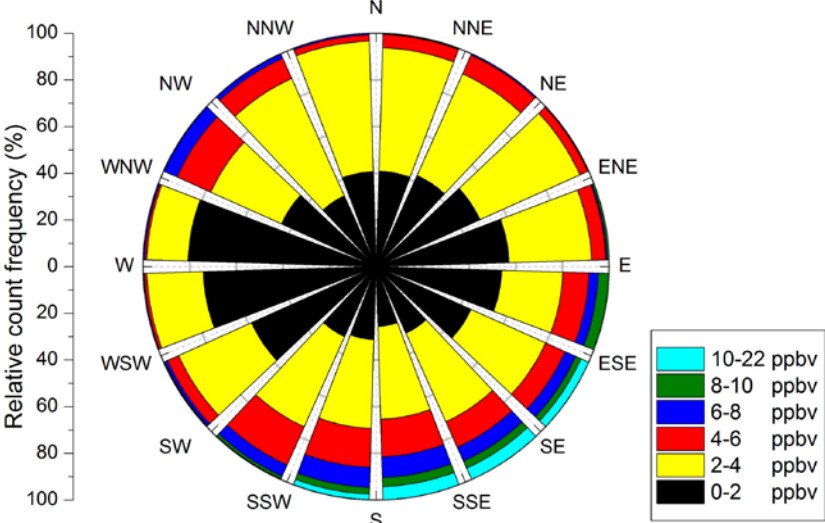

**Fig. 3.** Relative count frequency of NO$_2$ mixing ratios against wind direction. Mixing ratios higher than 10 ppbv were encountered only for wind directions between East-South-East and South-South-West, reflecting the impact of the local, major urban centres.



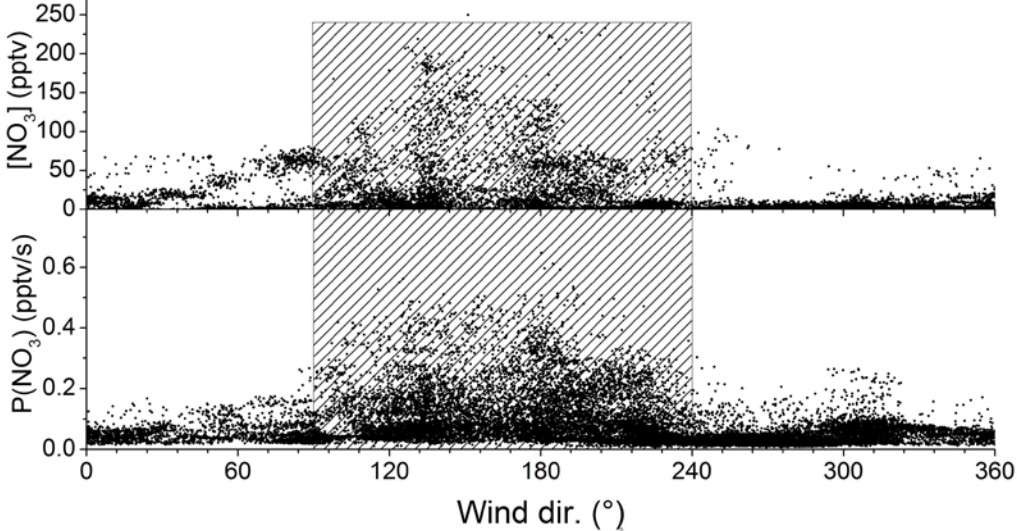

**Fig. 4.** [NO₃] and NO₃ production rate, P(NO₃) from R1, versus wind direction. The grey areas correspond to the Frankfurt-Mainz-Wiesbaden sector. Temperature dependent rate constants (NO₂ + O₃), used to calculate the NO₃ production term, were taken from an evaluation of kinetic data (Atkinson et al., 2004).



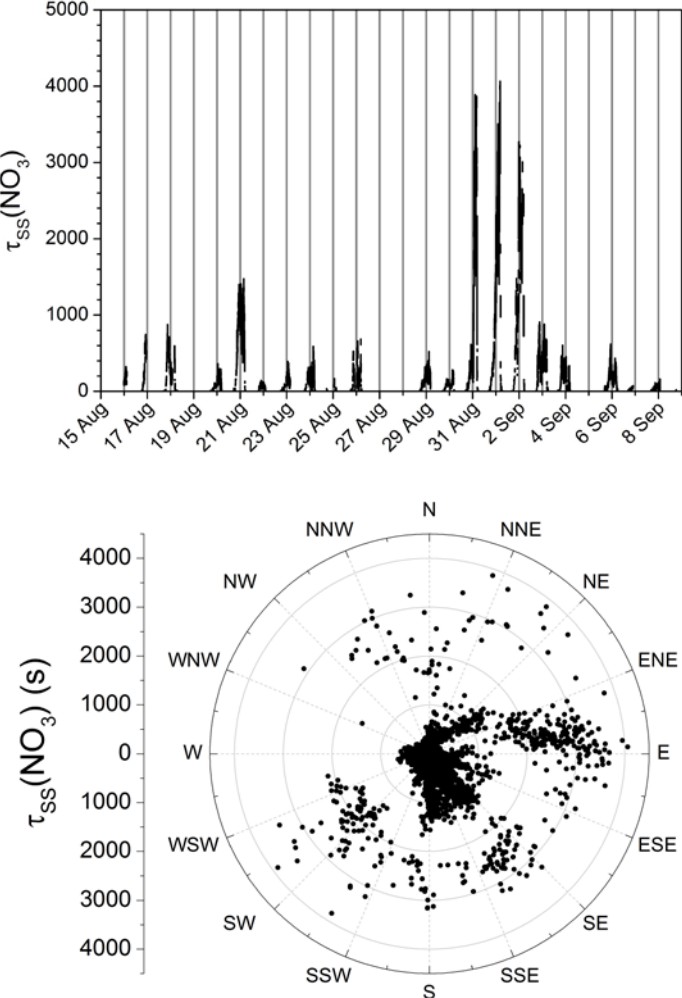

**Fig. 5.** *Upper panel*: $\tau_{ss}(NO_3)$ calculated according to Eq (3). *Lower panel*: $\tau_{ss}(NO_3)$ versus wind direction.





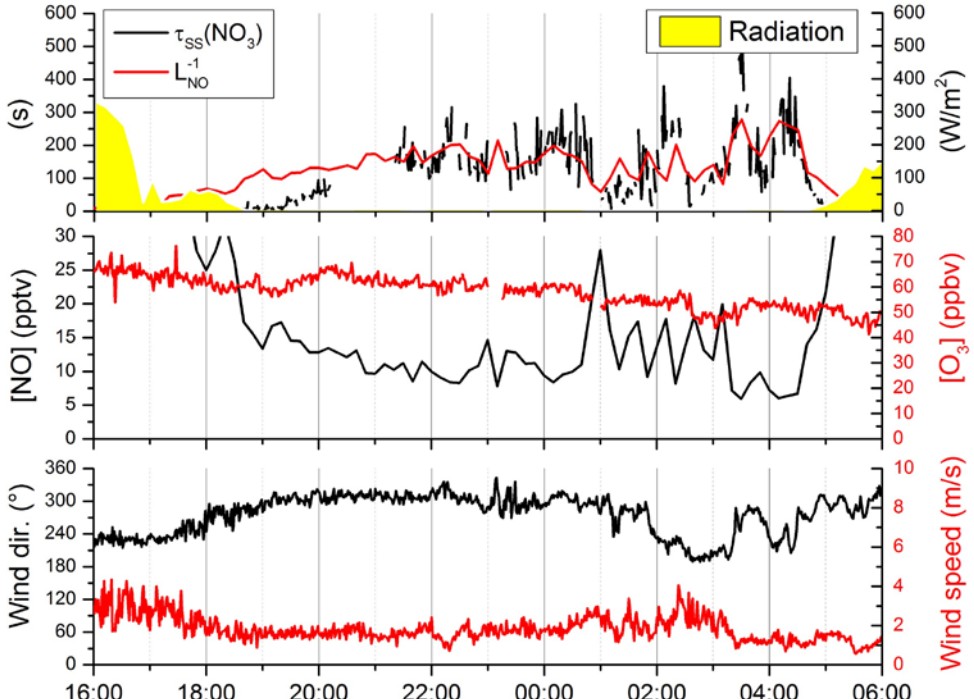

**Fig. 6.** *Upper panel*: $\tau_{ss}(NO_3)$ and inverse loss rate constant $(L_{NO})^{-1}$, for loss of $NO_3$ via reaction with NO during the night of the 23.08. The middle and lower panels display the associated [NO], [$O_3$] wind speed and wind direction data.





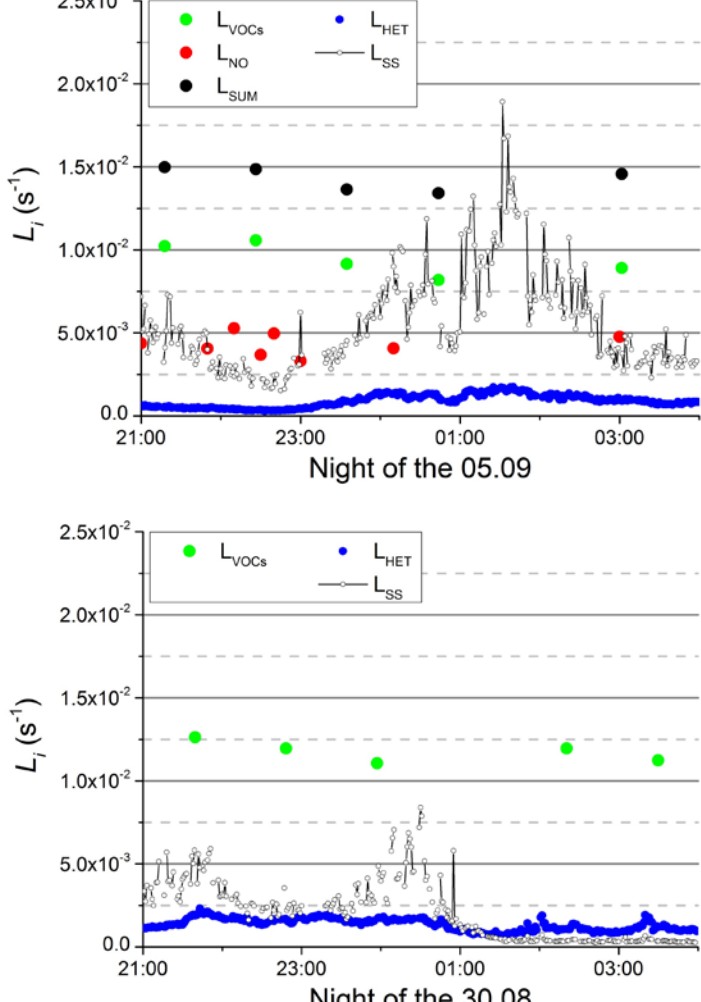

**Fig. 7.** *Upper panel*: Individual ($L_{VOCs} = \Sigma k_i[VOC]_i$, $L_{NO} = k_6.[NO]$ and $L_{HET}$) and summed $NO_3$ loss terms ($L_{SUM}$) for the night of the 05.09, together with the loss rate constant of $NO_3$ in steady-state ($L_{ss}$).
*Lower panel*: Individual ($L_{VOCs} = \Sigma k_i[VOC]_i$, $L_{NO} = k_6.[NO]$ and $L_{HET}$) and summed $NO_3$ loss terms ($L_{SUM}$) for the night of the 30.08, together with the inverse of the steady state lifetime of $NO_3$ ($L_{ss}$). $L_{VOCs}$, $L_{NO}$, $L_{HET}$ and $L_{ss}$ are given in the original time resolution (respectively 30min, 10min, 1min and 1 min). To calculate $L_{SUM}$, high resolution $L_{NO}$ and $L_{HET}$ were interpolated to the $L_{VOCs}$ time grid.





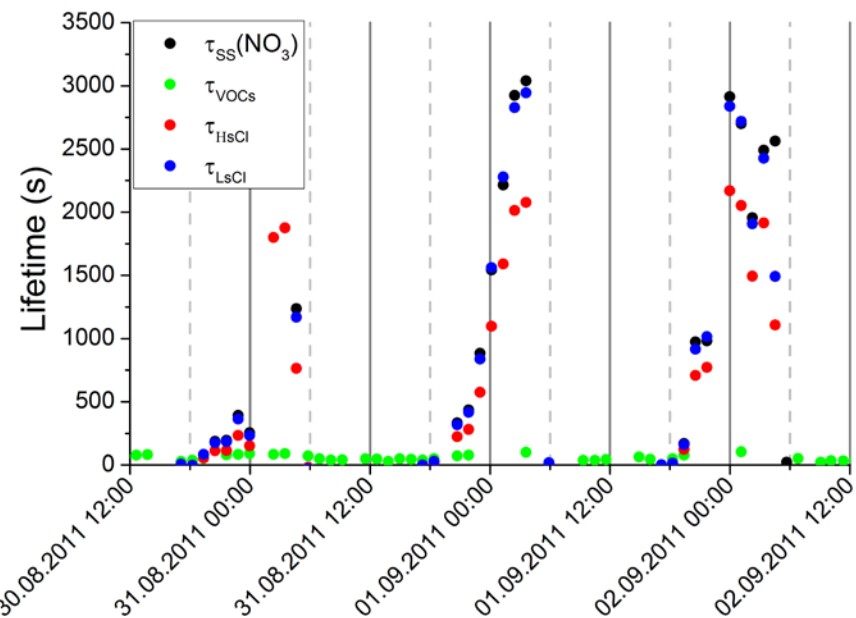

**Fig. 8.** $NO_3$ steady-state lifetimes calculated using different production terms. Black data points: Only $NO_2+O_3$ forms $NO_3$. Blue data points: additional formation from 0.01 pptv sCI reacting with $NO_2$. Red data points: additional formation from 0.1 pptv sCI reacting with $NO_2$. The inverse of $L_{VOCs}$ is given by the green data points. Similar to Fig. 7, all data were interpolated to the VOCs time axis hence the low time resolution.



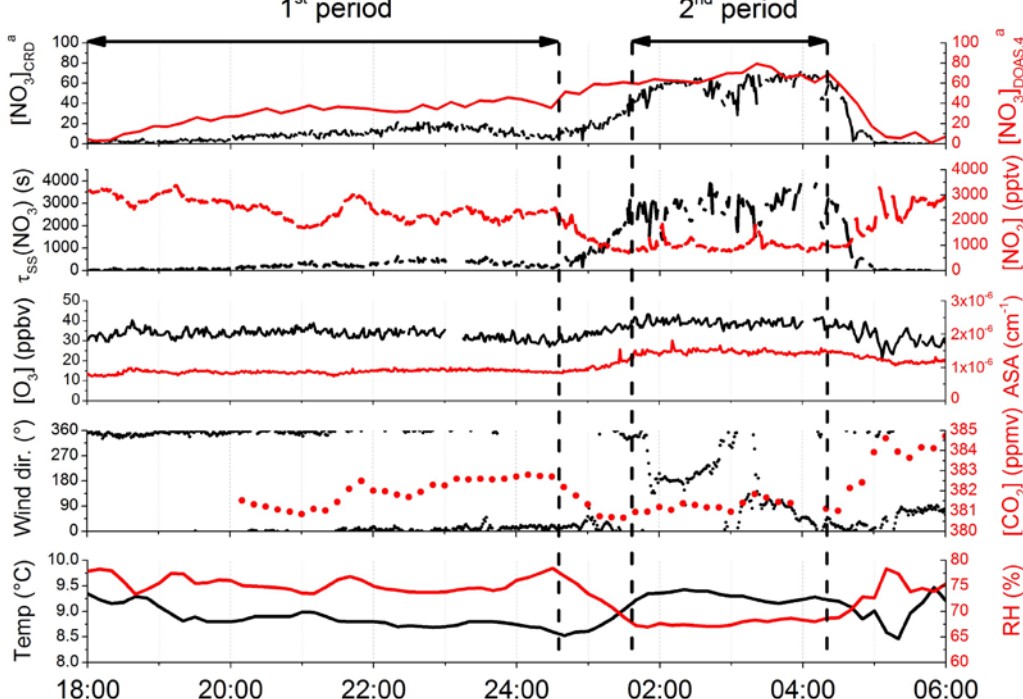

**Fig. 9.** Detailed view of the night of the 30.08 indicating two periods corresponding to different $NO_3$-lifetime regimes. The trends in $NO_3$ lifetime, $CO_2$, $NO_2$, $O_3$, RH, aerosol surface area (ASA) indicate a transition from boundary-layer to residual-layer air between ~00:30 and 01:30 in the morning. Sunrise was at about 05:00. [a]Units of pptv.




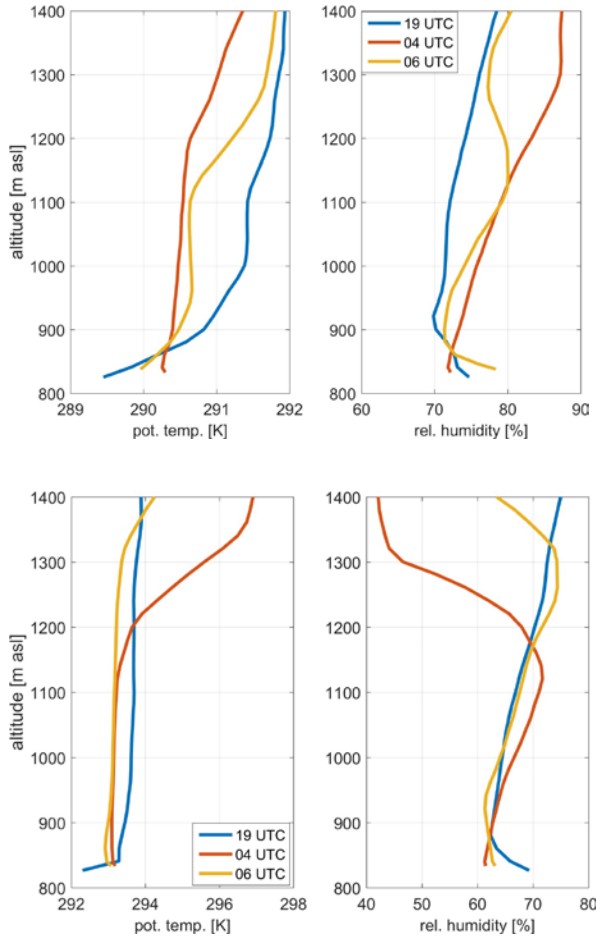

**Fig. 10.** *Upper panel:* Vertical profiles of the potential temperature (left) and the relative humidity (right) during the first period on 30.08 at 19:00 (blue) as shown in Fig. 9, and on 31.08 at 04:00 (red) and 06:00 (orange, after sunrise) during the second period.

 *Lower panel:* Same as above but for the 31.08 to 01.09.





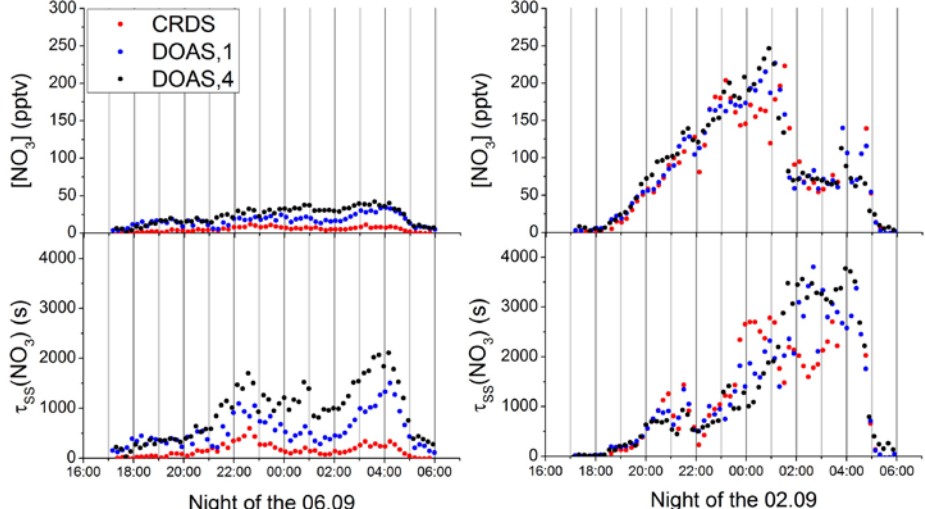

**Fig. 11.** *Left*: [NO$_3$] (top) and $\tau_{ss}$(NO$_3$) (lower panel) using CRDS data (red), lowest LP-DOAS light-path (DOAS,1) and highest LP-DOAS light-path (DOAS,4) for the night of the 06.09.

*Right:* as for the left-hand plot but showing data for the night of the 02.09.





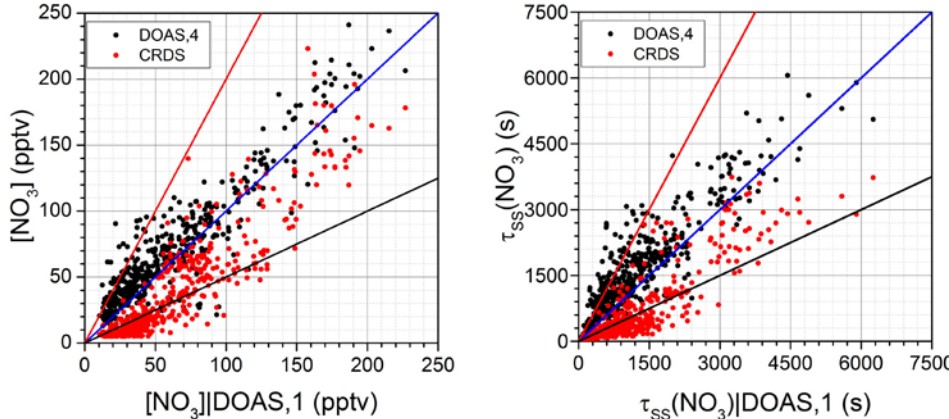

**Fig. 12.** *Left*: $[NO_3]_{CRDS}$ (red) and $[NO_3]_{DOAS,4}$ (black) versus $[NO_3]_{DOAS,1}$
*Right*: $\tau_{ss}(NO_3)_{CRDS}$ (red) and $\tau_{ss}(NO_3)_{DOAS,4}$ (black) versus $\tau_{ss}(NO_3)_{DOAS,1}$.
Coloured lines on both panels represent slopes of 0.5, 1 and 2 (black, blue and red, respectively).