# Peer review of "Chemical and meteorological influences on the lifetime of $NO_3$ at a semirural mountain site during "PARADE""

_Atmospheric Chemistry and Physics, 2016_

## Referee Comment (RC1) · S. Brown (Referee) · 26 Feb 2016

General Comments:

This paper analyzes the lifetimes of NO3 and N2O5 based on measurements from a mountaintop site in west central Germany during late summer 2011. The data include both in-situ measurements from a CRDS instrument and vertically resolved measurements from a long path DOAS instrument. Additional supporting measurements of NOx, O3, aerosols and VOCs are available for a reasonably complete assessment of both the production and loss processes for the NO3 and N2O5 reactive intermediates. A key observation is that the steady state lifetimes for NO3 are at times considerably greater than would be calculated from the loss of NO3 to VOCs, particularly biogenics.

[Figure]

The attempt to explain this observation leads to two key results.

The first is a consideration of the potential $NO_3$ source from the oxidation of $NO_2$ by stabilized Criegee intermediates (sCI) produced in the ozonolysis of alkenes. This production has not to my knowledge been considered in the context of field data previously. Although the effect is not large enough to explain the observed discrepancies in $NO_3$ lifetimes, the thorough estimations of the range of sCI concentrations and associated rate coefficients for reactions with $NO_2$ are a highly valuable contribution to the literature. The analysis shows in particular that there may be situations where this reaction is non-negligible in comparison to the conventional $NO_3$ source from the reaction of $NO_2$ with $O_3$.

The second is an analysis of mixing effects between different altitudes within the nocturnal boundary layer / residual layer structure. The combination of CRDS and DOAS measurements provides a convincing data set to demonstrate the important role that mixing can play in determining $NO_3$ concentrations and apparent lifetimes. While these effects have been considered previously (e.g., Stutz 2004), the analysis here provides a compelling example.

The paper is a valuable contribution to the literature and should be published in ACP subject to consideration of the list of minor comments below.

Specific Comments:

Page 4, line 4. Site altitude is given in meters above sea level. What is the altitude of the surrounding area? How far above ground level is this site? This is important in understanding whether the site should be in the broader nocturnal boundary layer or the residual layer.

Page 4, line 32. Suggest using a different description than "excellent" for the instrument agreement, since 10% is not especially good agreement for $O_3$.

Page 5, line 26. "Ethane" is repeated. Possibly "ethene" is meant?

Line 6, section 2.25. Were particle surface areas measured at dry or ambient RH, or at some intermediate RH, and was there a need to correct surface areas for hygroscopic growth?

Page 8, line 3-6. A more traditional wind rose is described in the text but not shown in figure 3. This would be helpful alongside the existing figure 3.

Page 8, line 8-9. Statement that air masses from local urban area are expected to have higher aerosol and VOC implies that such measurements were not available. Suggest removing the word "expected" in favor of a quantitative statement about the wind direction dependence of the available aerosol and VOC data.

Page 8, line 12. Suggest inserting the word "likely" since there is no quantitative assessment of the droplet surface area or heterogenous uptake rates. Could also include a standard citation for this effect if desired (e.g., Leliveld & Crtuzen, 1990).

Page 10, line 28-30. Statement is not consistent. The 2 hour approach to steady state cited in section 3.3 and figure S3 uses LVOCs (here equivalent to LNO) of 2.5e-3 s-1, not 0.01 s-1 as shown in figure 6. Disagreement between calculated and observed NO3 lifetimes would seem to be less likely due to lack of steady state than shown by the supplemental figure.

Page 11, line 6. A gamma value of 0.02 may be consistent with recommendations but lies on the high end of the range of the values determined in both laboratory and field studies, which show values as low as 1e-4 (e.g., Brown & Stutz, Chem. Soc. Rev., 41, 6405-6447, 2012). The comparison between the calculated heterogeneous loss and the observed lifetimes thus represents something close to an upper limit of the former to the latter. Lower gamma would explain why heterogeneous loss sometimes exceeds the calculated lifetimes. The possibility of much lower uptake coefficients should be referenced in this discussion.

Page 13, line 4-8. The factor of 5-6 variability in tau(NO3) is an overstatement of

the actual variability in the loss of the sum of NO3 and N2O5, as indicated by the concurrent change in tau(N2O5). What is the variability in P(NO3) during this time?

Page 14, line 9 and following discussion. What is meant by the term "low-lying residual layer". The measurement site is already at some altitude above ground level (see comment above) and may already be above what would normally be considered the height of the nocturnal boundary layer. Is this mountaintop site not already in the residual layer, or are the local boundary layer dynamics influenced by the terrain so as to create some sort of local nocturnal boundary layer effect?

Page 14, line 28. Should "above the residual layer" be "above the nocturnal boundary layer" ?

Page 15, line 32-33. Could the gradient be due to downward mixing of NO3 or N2O5 from altitudes not sampled by the DOAS – i.e., presumably the measured gradient does not end at the highest DOAS level, so the observed change could be due more to mixing than to variation in the loss term.

Page 16, lines 21-29. The argument regarding local BVOC emissions oxidized by NO3 within a gradient is quite plausible and likely the best explanation of the observations. However, should it also be accompanied by a shift in the equilibrium ratio of N2O5 to NO3? In other words, if the NO3 were lost to BVOC reactions near the observatory, the N2O5 to NO3 ratio would possibly not respond as quickly, leading to a larger observed ratio of N2O5 to NO3 than that predicted by equilibrium. Can the authors comment on the potential magnitude of this effect, for example using the NO3-BVOC rate constant compared to the N2O5 thermal dissociation rate constant? If so, is there any evidence from the N2O5 to NO3 ratio so support this hypothesis?

---

## Referee Comment (RC2) · Anonymous Referee #2 · 26 Feb 2016

General comments:

This well written manuscript by Sobanski et al. outlines a comprehensive study of NO3 mixing ratios and lifetimes in the context of a field campaign (referred to by the acronym PARADE) in the German mountain region, Taunus, during Summer 2011. The campaign was equipped with a broad range of instrumentation to detect most atmospheric species (as well as meteorological data) that are relevant for NO3 generation and destruction in order to interpret the key findings, which included unusually large and also highly variable NO3 mixing ratios, and long lifetimes (up to 1 hr and more). The general discussion uses a steady state model which is based on the most relevant NO3 production reaction at nighttime (NO2 + O3). The manuscript also considered nonesteady state conditions and contains novel aspects such as the influence of Criegee intermediates on the NO3 production rates through reactions with NO2; also the reaction of OH and HNO3 was considered. Loss mechanisms of NO3 are discussed based on reaction with NO and volatile organic compounds (VOC, biogenic and anthropogenic). Data are additionally interpreted on basis of meteorological conditions and in this context it is argued that the high NO3 concentration on some occasions are likely to be due to a "low lying residual layer" over the boundary layer with a significant positive NO3 concentration altitude gradient.

Key instruments are (i) a cavity ring down spectrometer (CRDS) for the detection of NO3 and N2O5 (in a heated channel) as well as (ii) a 4-beam long-path differential optical absorption spectrometer (LP-DOAS). A comparison of the corresponding data is also included in the paper mostly to argue for the residual layer hypothesis.

Generally, the manuscript is well structured and there is a good flow of the text. The discussion of data dwells on the most significant events and atmospheric scenarios during the 3 week campaign and the items that require discussion and explanation have been carefully selected. Based on the methodology the data inspire confidence and the majority of conclusions are supported by arguments that are anchored in the data, however some discrepancies remain.

This manuscript clearly merits publication in ACP, however, it is recommended that the comments and minor corrections listed below are considered prior to acceptance.

Specific comments and observations:

Section 1: Page 1&2: The introduction is giving a good overview of the most relevant atmospheric reactions that influence the NO3 concentration. This part of the manuscript would however benefit from more quantitative data on the reaction rates as they are known in the literature (IUPAC and Atkinson et al.). Reaction rates could be incorporated in the R-equations or presented in form of a table and would provide the reader with kinetic information on the relevance of each reaction for the NO3 chemistry.

Section 2: The information on the instrumentation specification is brief, as most of the experimental setup utilized during PARADE have been published before. However, the reader would benefit from some more information on the key instruments. Page 5: What zeroing/calibration procedures for the CRDS instruments were applied? Page 5: The description of the NO2 instrument (Page 5) is particularly short (Thieser et al. 2015).

Page 5 & 6: The literature used for the cross-sections for NO3 and NO2 does not seem to be mentioned (neither for the CRDS nor the LP-DOAS setups).

Page 5, Line 24: What does "quasi online" mean?

Page 6: An overall uncertainty of the LP-DOAS setup of 10% for NO3 is stated. The cross-section uncertainty is of that order. The error evaluation seems to be too optimistic.

Section 3: Page 9: Please check (Eq4). If L is a loss rate constant in units of [s-1] the power of "−1" seems out of place.

Page 9, Line24: What does the index 'DER' stand for?

Page 9, Line30: Why is it meaningful to state an average value of the steady state lifetime, if the night to night variability is high? What is meant by values (plural)?

Page 10, Line 8: "Because the error made by assuming steady state is generally small ...", can this be quantified?

Page 11, Line 26: What is meant by "weaker day/night cycle"?

Page 12, Line25: What is meant by "these time scales"?

Page 13: The arguments in Section 3.3.4 are well laid out and appear conclusive; nevertheless the section would benefit from explicit mentioning of reaction rates (e.g. for sCI +NO2; also see comment above).

Page 13 (bottom part): Even though sCI mixing ratios of ∼4 pptv seem clearly too high, the authors outline an interesting hypothesis that merits further research.

Page 14: What is meant by "low-lying residual layer"? What constitutes this layer and how is it contrasted to the boundary layer, as the measurements were taken at some height of 825 m ASL.

Page 15: Fig. S2 in the main text would be really helpful for the discussion of the observed NO3 vertical concentration gradient. The altitudes of the sender/receiver units and retro-reflectors of the LP-DOAS setups in comparison to the altitude of the CRDS instrument should be repeated here for the benefit of the reader.

Page 15: Is there any evidence for mixing of the layers when comparing the four DOAS directions and different heights probed during the measurements?

Page 16, Line 16: How can be assumed that there are no significant NO3 gradients in the proposed "residual layer". Again, is there evidence for mixing owing to convective currents in the time series of the data on the relevant days?

Page 16, Line7-9: Does Fig 12 (left panel) really allow this statement? Can the statement be underpinned by quantitative information on altitude dependence, and the term "closer" through R-values for example.

Page 16: The last paragraph in section 3.3.6 seems to be better placed in the previous section on the discussion of the residual layer and not in the comparison of CRDS and LP-DOAS.

Generally, some material from the supplementary material would have been helpful in the main text; notably Fig S1, S2, S4 and S8. S1 and S2 are important to get an idea of the topography in the vicinity of the measurement site. Especially in the context of section 3.3.5 and 3.3.6 these figures are helpful.

Technical corrections:

In many places mixing ratios are not expressed by volume. Examples are: P5,L23,L26,L27; P8, L15;P14,L19; P16,L4,L11.

Page 3, Line 23: State the month(s) in 2011 explicitly here.

Page 4, Line 13: "Fig. S1" should be highlighted.

Page 6, Line 7: "broadbanding"? Is that the term to use? Moreover, the information in this line is redundant and was mentioned before on page 3.

Page 9, L26: In Eq 5 the minus signs in the denominator are hardly discernible.

Page 11, Line 6: IUPAC should get a reference.

Page 11, Line 29: "Fig. 7" should be highlighted.

Fig. 1 & 2: There is a discrepancy on the third cold front arrival time (third blue vertical line). It may be better to merge Figs. 1 & 2 into one figure with six panels. If this could be done without loss of plot quality, this would help the reader.

Fig. 5 Units of seconds are missing on the vertical axis of the upper panel.

Fig 6. The red and black trace refer to the left axis and radiation (in yellow) to the right axis. This is confusing, as the other panels are colour coded.

Fig 7. The lower panel does not seem to show LSUM, LNO. The stated time grid appears inconsistent with the dots in the figure (e.g. 30 min for LVOCS).

Page 22, Lines 2&3: "...of 4 the..." and "...at the 5 Großer Feldberg."?

---

## Author Comment (AC1) · 11 Mar 2016

We thank Steven Brown for these constructive and helpful comments. Our replies (in blue/red) to each comment (in black) are listed below. Red text indicates changes to the manuscript.

| Referee 1 (Steven Brown) |
|---|
| *General Comment* |
| This paper analyzes the lifetimes of NO3 and N2O5 based on measurements from a mountaintop site in west central Germany during late summer 2011. The data include both in-situ measurements from a CRDS instrument and vertically resolved measurements from a long path DOAS instrument. Additional supporting measurements of NOx, O3, aerosols and VOCs are available for a reasonably complete assessment of both the production and loss processes for the NO3 and N2O5 reactive intermediates. A key observation is that the steady state lifetimes for NO3 are at times considerably greater than would be calculated from the loss of NO3 to VOCs, particularly biogenics.

The attempt to explain this observation leads to two key results. The first is a consideration of the potential NO3 source from the oxidation of NO2 by stabilized Criegee intermediates (sCI) produced in the ozonolysis of alkenes. This production has not to my knowledge been considered in the context of field data previously. Although the effect is not large enough to explain the observed discrepancies in $NO_3$ lifetimes, the thorough estimations of the range of sCI concentrations and associated rate coefficients for reactions with NO2 are a highly valuable contribution to the literature. The analysis shows in particular that there may be situations where this reaction is non-negligible in comparison to the conventional NO3 source from the reaction of NO2 with O3.

The second is an analysis of mixing effects between different altitudes within the nocturnal boundary layer / residual layer structure. The combination of CRDS and DOAS measurements provides a convincing data set to demonstrate the important role that mixing can play in determining NO3 concentrations and apparent lifetimes. While these effects have been considered previously (e.g., Stutz 2004), the analysis here provides a compelling example.

The paper is a valuable contribution to the literature and should be published in ACP subject to consideration of the list of minor comments below. |
| We thank the reviewer for this positive assessment of our manuscript. |
| *Specific Comments* |
| Page 4, line 4. Site altitude is given in meters above sea level. What is the altitude of the surrounding area? How far above ground level is this site? This is important in understanding whether the site should be in the broader nocturnal boundary layer or the residual layer. |
| As the Kleiner Feldberg is part of a mountain range, the altitude of the surrounding area depends on direction and distance from the summit.. The closest cities are at altitudes of about $120 \pm 50$ m. We add this information on page 4 and refer to a previous publication (Crowley et al, 2010) in which a topographical map was published. |
| Page 4, line 32. Suggest using a different description than "excellent" for the instrument agreement, since 10% is not especially good agreement for O3. |
| There were, in total, six measurements of $O_3$. Some were research instruments and some were monitoring instruments with infrequent calibration and this may be the cause of some "disagreement". We now suggest that the agreement was good, rather than |

| |
|---|
| excellent. |
| Page 5, line 26. "Ethane" is repeated. Possibly "ethene" is meant? |
| Yes, it should have been ethene. Corrected |
| Page 6, section 2.25. Were particle surface areas measured at dry or ambient RH, or at some intermediate RH, and was there a need to correct surface areas for hygroscopic growth? |
| We have added the text: Aerosol was sampled at ambient RH and no correction was applied for hygroscopic growth. |
| Page 8, line 3-6. A more traditional wind rose is described in the text but not shown in figure 3. This would be helpful alongside the existing figure 3. |
| Wind rose added in Figure 3 |
| Page 8, line 8-9. Statement that air masses f rom local urban area are expected to have higher aerosol and VOC implies that such measurements were not available. Suggest removing the word "expected" in favor of a quantitative statement about the wind direction dependence of the available aerosol and VOC data. |
| As expected, the data show that the high $NO_x$ from the urban sector was associated with the largest mixing ratios of anthrogogenic hydrocarbons. We now write:
High $NO_x$ levels measured in this sector were associated with the highest levels of anthropogenic hydrocarbons such as e.g. toluene. |
| Page 8, line 12. Suggest inserting the word "likely" since there is no quantitative assessment of the droplet surface area or heterogenous uptake rates. Could also include a standard citation for this effect if desired (e.g., Leliveld & Crtuzen, 1990). |
| We now write: Precipitation periods were associated with $[NO_3]$ and $[N_2O_5]$ below the detection limit, which is likely a result of $N_2O_5$ uptake to droplets. |
| Page 10, line 28-30. Statement is not consistent. The 2 hour approach to steady state cited in section 3.3 and figure S3 uses LVOCs (here equivalent to LNO) of 2.5e-3 s-1, not 0.01 s-1 as shown in figure 6. Disagreement between calculated and observed NO3 lifetimes would seem to be less likely due to lack of steady state than shown by the supplemental figure. |
| This is true. We now write: During the period from 19:00 to 21:00 the lifetime of $NO_3$ attributable to reaction with NO, $\tau(NO)$, is greater than $\tau_{ss}(NO_3)$, which may be due both to non-acquisition of steady state (for the first hour or so) and a contribution of other loss processes. |
| Page 11, line 6. A gamma value of 0.02 may be consistent with recommendations but lies on the high end of the range of the values determined in both laboratory and field studies, which show values as low as 1e-4 (e.g., Brown & Stutz, Chem. Soc. Rev., 41, 6405-6447, 2012). The comparison between the calculated heterogeneous loss and the observed lifetimes thus represents something close to an upper limit of the former to the latter. Lower gamma would explain why heterogeneous loss sometimes exceeds the calculated lifetimes. The possibility of much lower uptake coefficients should be referenced in this discussion. |
| We agree. The text now reads: This expression is based on laboratory studies and results in a value at 283 K of $\sim 2 \times 10^{-2}$ which is larger than uptake coefficients obtained by analysis of both laboratory and field data in which the particle is not purely inorganic (Brown and Stutz, 2012). During periods when the $NO_3$ lifetime is long (e.g. $\tau_{ss}(NO_3) > 2000$ s or $L_{ss}(NO_3) < 5 \times 10^{-4}$ s$^{-1}$) as on the second period of the night of the 30.08 (lower panel), when using the $\gamma$ parameterisation above, the losses of $N_2O_5$ to particles can account entirely for the observed lifetimes, or even exceed the steady state reactivity, which may be the result of using a high value for $\gamma$. |

Page 14, line 4-8. The factor of 5-6 variability in tau(NO3) is an overstatement of the actual variability in the loss of the sum of NO3 and N2O5, as indicated by the concurrent change in tau(N2O5). What is the variability in P(NO3) during this time?

The sentence was misleading and should have referred to the gradient in the NO3 lifetime rather than its variability. We now write:

Close examination of the $NO_3$ lifetime on these nights reveals an especially strong gradient, with $\tau_{ss}(NO_3)$ values increasing by e.g. 500 to 600 % over the course of an hour. Simultaneously, $\tau_{ss}(N_2O_5)$ values (not shown) increase by 200 %. The smaller increase in $\tau_{ss}(N_2O_5)$ is due to a simultaneous decrease of $[NO_2]$ by 50 % thus shifting the equilibrium towards $NO_3$. The large gradient in $\tau_{ss}(NO_3)$ cannot be attributed to a decrease in the mixing ratios of reactive trace gases, which were much less variable (see above). We therefore consider the possibility that the long lifetimes observed on these nights are related to sampling air masses from a low-lying residual layer (i.e. from above a particulary shallow nocturnal boundary layer) which, especially at low wind speeds, is decoupled from ground emissions of NOx and VOCs, allowing $NO_3$ and $N_2O_5$ to build up to higher levels and would also help to explain explain the lack of dependence on wind direction.

Page 14, line 9 and following discussion. What is meant by the term "low-lying residual layer". The measurement site is already at some altitude above ground level (see comment above) and may already be above what would normally be considered the height of the nocturnal boundary layer. Is this mountaintop site not already in the residual layer, or are the local boundary layer dynamics influenced by the terrain so as to create some sort of local nocturnal boundary layer effect?

The height difference between the summit of the mountain and the surrounding, countryside and cities is about 700m, i.e. higher than the nocturnal boundary layer of the surrounding areas. As the reviewer states, the NBL will follow, to some extent the topography so that a variable height NBL may be expected and which will depend on wind-flow and terrain structure. Essentially, the "low-lying residual layer" is a consequence of a very shallow NBL. We now qualify the statement related to sampling air from a low-lying residual layer in the following manner:

We therefore consider the possibility that the long lifetimes observed on these nights are related to sampling air masses from a low-lying residual layer (i.e. from above a particulary shallow nocturnal boundary layer) which, especially at low wind speeds, is decoupled from ground emissions of NOx and VOCs, allowing $NO_3$ and $N_2O_5$ to build up to higher levels and would also help to explain explain the lack of dependence on wind direction.

Page 14, line 28. Should "above the residual layer" be "above the nocturnal boundary layer" ?

Yes. This has been corrected.

Page 15, line 32-33. Could the gradient be due to downward mixing of NO3 or $N_2O_5$ from altitudes not sampled by the DOAS – i.e., presumably the measured gradient does not end at the highest DOAS level, so the observed change could be due more to mixing than to variation in the loss term.

This is true. However, as the NO3 production term does not change with height, changes in concentration are still related to the loss term for the sampled air-mass irrespective of whether the air sampled at any one DOAS light-path is influenced by mixing from above or not. i.e. if NO3 exists at further elevated concentrations above the highest DOAS light path, this is still a result of an extended lifetime aloft.

Page 16, lines 21-29. The argument regarding local BVOC emissions oxidized by $NO_3$ within a gradient is quite plausible and likely the best explanation of the observations.

However, should it also be accompanied by a shift in the equilibrium ratio of N2O5 to NO3? In other words, if the NO3 were lost to BVOC reactions near the observatory, the N2O5 to NO3 ratio would possibly not respond as quickly, leading to a larger observed ratio of N2O5 to NO3 than that predicted by equilibrium. Can the authors comment on the potential magnitude of this effect, for example using the NO3-BVOC rate constant compared to the N2O5 thermal dissociation rate constant? If so, is there any evidence from the N2O5 to NO3 ratio so support this hypothesis?

There was no evidence of a significant reduction in NO3 to N2O5 ratio. However, the time required for NO2, NO3 and N2O5 to acquire equilibrium is the inverse sum of the forward and back, first-order rate constants i.e. 1/(kb + kf[NO2]). At 800 mbar and 285, kb is ~0.008 s-1 and kf ~ 1.2e-12. Taking a low value of [NO2] of 1 ppb then results in a relaxation time (to equilibrium) of 10 s. i.e. equilibrium is reached on a much shorter time scale than steady-state and will generally hold unless NO3 is lost very locally (e.g. in the inlet) or at low NO2 and low temperatures.

---

## Author Comment (AC2) · 11 Mar 2016

We thank the reviewer for these constructive and helpful comments. Our replies (in blue/red) to each comment (in black) are listed below. Red text indicates changes to the manuscript.

| Referee 2 |
|---|
| *General Comment* |
| This well written manuscript by Sobanski et al. outlines a comprehensive study of NO3 mixing ratios and lifetimes in the context of a field campaign (referred to by the acronym PARADE) in the German mountain region, Taunus, during Summer 2011. The campaign was equipped with a broad range of instrumentation to detect most atmospheric species (as well as meteorological data) that are relevant for NO3 generation and destruction in order to interpret the key findings, which included unusually large and also highly variable NO3 mixing ratios, and long lifetimes (up to 1 hr and more). The general discussion uses a steady state model which is based on the most relevant NO3 production reaction at nighttime (NO2 + O3). The manuscript also considered none-steady state conditions and contains novel aspects such as the influence of Criegee intermediates on the NO3 production rates through reactions with NO2; also the reaction of OH and HNO3 was considered. Loss mechanisms of NO3 are discussed based on reaction with NO and volatile organic compounds (VOC, biogenic and anthropogenic). Data are additionally interpreted on basis of meteorological conditions and in this context it is argued that the high NO3 concentration on some occasions are likely to be due to a "low lying residual layer" over the boundary layer with a significant positive NO3 concentration altitude gradient. Key instruments are (i) a cavity ring down spectrometer (CRDS) for the detection of NO3 and N2O5 (in a heated channel) as well as (ii) a 4-beam long-path differential optical absorption spectrometer (LP-DOAS). A comparison of the corresponding data is also included in the paper mostly to argue for the residual layer hypothesis. Generally, the manuscript is well structured and there is a good flow of the text. The discussion of data dwells on the most significant events and discussion of data dwells on the most significant events and atmospheric scenarios during the 3 week campaign and the items that require discussion and explanation have been carefully selected. Based on the methodology the data inspire confidence and the majority of conclusions are supported by arguments that are anchored in the data, however some discrepancies remain. |
| This manuscript clearly merits publication in ACP, however, it is recommended that the comments and minor corrections listed below are considered prior to acceptance. |
| We than the reviewer for this positive assessment of our manuscript. |
| *Specific Comments* |
| Section 1: Page 1&2: The introduction is giving a good overview of the most relevant atmospheric reactions that influence the NO3 concentration. This part of the manuscript would however benefit from more quantitative data on the reaction rates as they are known in the literature (IUPAC and Atkinson et al.). Reaction rates could be incorporated in the R-equations or presented in form of a table and would provide the reader with kinetic information on the relevance of each reaction for the NO3 chemistry. |
| The discussion at this point is largely qualitative. Later in the manuscript we discuss rates of production and loss of NO$_3$ in quantitative terms and also list rate coefficients in a Table. We prefer not to reduce the readability of this section by introduction of reaction rate constants. |
| Section 2: The information on the instrumentation specification is brief, as most of the experimental setup utilized during PARADE have been published before. However, the |

reader would benefit from some more information on the key instruments. Page 5: What zeroing/calibration procedures for the CRDS instruments were applied? Page 5: The description of the NO2 instrument (Page 5) is particularly short (Thieser et al. 2015).

We have added text to describe how the CRDS measuring $NO_3$ and $N_2O_5$ was zeroed: Zeroing was performed by addition of NO (R6).

We have extended the information on the $NO_2$ instrument:

The instrument was zeroed using dry, synthetic air and corrections were made for humidity differences between the zero air and ambient. Typical ring-down times were ~ 30 µs and the instrument has a detection limit for $NO_2$ of ≈ 20 pptv in 1 min and an accuracy of 6 % where the dominant contribution is uncertainty in the $NO_2$ cross sections (Voigt et al., 2002).

Page 5 & 6: The literature used for the cross-sections for NO3 and NO2 does not seem to be mentioned (neither for the CRDS nor the LP-DOAS setups).

We now write (for $NO_3$) : The total uncertainty was ~15 % for both $NO_3$ and $N_2O_5$, with the largest contributions arising from uncertainty in the $NO_3$ cross section (Orphal et al., 2003; Osthoff et al., 2007) and $NO_3$ losses.
The source of the $NO_2$ cross sections is now also listed (see reply to comment above).

Page 5, Line 24: What does "quasi online" mean?

The term "quasi-online" was unnecessary and has been removed

Page 6: An overall uncertainty of the LP-DOAS setup of 10% for NO3 is stated. The cross-section uncertainty is of that order. The error evaluation seems to be too optimistic.

The NASA evaluation panel recommends a value of $2.25 \pm 0.15 \times 10^{-17}$ cm$^2$ molecule$^{-1}$ for the cross section at 662 nm, i.e. errors of 6.6 %.

Section 3: Page 9: Please check (Eq4). If L is a loss rate constant in units of [s-1] the power of "–1" seems out of place.

The comment is correct. An early version of this expression referred to lifetimes, hence the power term. This has been corrected.

Page 9, Line24: What does the index 'DER' stand for?

To clarify this we now write: The lifetime calculated via derivatives, $\tau_{DER}$ ($NO_3$) is given by………

Page 9, Line30: Why is it meaningful to state an average value of the steady state lifetime, if the night to night variability is high? What is meant by values (plural)?

We give an average mixing ratio to highlight the fact that so that nights with lifetimes of e.g. >1000 s are to exceptional. The use of a plural was an error and has been corrected.

Page 10, Line 8: "Because the error made by assuming steady state is generally small . . .", can this be quantified?

Yes. We now write: The derivatives method does not provide data for the whole campaign. $\tau_{DER}(NO_3)$ are sometime very scattered and sometimes negative which arises from noise on the derivative terms. However, on average, the ratio $\tau_{ss}(NO_3)$ / $\tau_{DER}(NO_3)$ was 0.99 with ~66 % of the lifetimes agreeing within 30 %. The the analysis and discussion below is based on the lifetimes obtained by the steady-state analysis.

Page 11, Line 26: What is meant by "weaker day/night cycle"?

We now use the term "weaker diel variation"

Page 12, Line25: What is meant by "these time scales"?

The term "these time scales" was superfluous and has been removed.

Page 13: The arguments in Section 3.3.4 are well laid out and appear conclusive; nevertheless the section would benefit from explicit mentioning of reaction rates (e.g. for sCI +NO2; also see comment above).

Specific literature measurements of the rate constants for sCI + $NO_2$ are mentioned in the text and an "average" value listed in Table 3. We consider this sufficient.

Page 13 (bottom part): Even though sCI mixing ratios of 4 pptv seem clearly too high, the authors outline an interesting hypothesis that merits further research.

We agree. We have little information on the concentration of sCI in the atmosphere and their reactions.

Page 14: What is meant by "low-lying residual layer"? What constitutes this layer and how is it contrasted to the boundary layer, as the measurements were taken at some height of 825 m ASL.

See also reply to the similar comment of reviewer 1. The "low-lying residual layer" is essentially a very shallow boundary layer. We now write:

In summary, the data shown in Fig. 9 and the LP-DOAS measurements of $NO_3$ at different altitudes provide compelling evidence for a low-lying residual layer (or very shallow nocturnal boundary layer)

Page 15: Fig. S2 in the main text would be really helpful for the discussion of the observed NO3 vertical concentration gradient. The altitudes of the sender/receiver units and retro-reflectors of the LP-DOAS setups in comparison to the altitude of the CRDS instrument should be repeated here for the benefit of the reader.

Done. We now write:

In Fig. 12 (left panel) we plot the campaign $NO_3$ mixing ratios measured by the CRDS and DOAS,4 versus the DOAS,1 dataset. As mentioned previously and illustrated in Fig S2, the LP-DOAS light source and the CRDS inlet were co-located at a heigth of ~835 m whilst the LP-DOAS retro-reflectors were at ~959 and 872 m.

Page 15: Is there any evidence for mixing of the layers when comparing the four DOAS directions and different heights probed during the measurements?

We cannot derive information about mixing from the DOAS measurements alone. To clarify this we now write (page 17): The DOAS measurement of concentration at different altitudes does not contain information about the vertical exchange and mixing itself. Any estimation of vertical mixing for a reactive trace-gas like $NO_3$ would require a chemical model with transport, which would be difficult for such an environment with complex topography like the Kleiner Feldberg.

Page 16, Line 16: How can be assumed that there are no significant NO3 gradients in the proposed "residual layer". Again, is there evidence for mixing owing to convective currents in the time series of the data on the relevant days?

The use of the term "no significant gradient" was misleading. We now write: The apparent dependence of the $NO_3$ mixing ratios and lifetime on the $NO_3$ concentrations reflects the fact that, when $NO_3$ is high, both instruments are sampling the residual layer in which $NO_3$ levels are expected to be higher and gradients in $NO_3$ are exected to be weaker than found close to the surface (Brown et al., 2007).

Page 16, Line7-9: Does Fig 12 (left panel) really allow this statement? Can the statement be underpinned by quantitative information on altitude dependence, and the term "closer" through R-values for example.

The Figure shows that at low $NO_3$ mixing ratios, the ratio of CRDS to DOAS,1 is nearly always below 1, where as that of DOAS,4 to DOAS,1 is nearly always greater than one. This is very strong evidence of a gradient. When $NO_3$ mixing ratios are high (above 100 pptv) the gradient is weaker. We have added text to illustrate this in a more quantitative manner:

To illustrate this, For $NO_3$ steady state lifetimes of < 1500 (i.e. all nights except for 31.08-03.09)  s the average value of  the $[NO_3]_{DOAS,4}$ / $[NO_3]_{DOAS,1}$ ratio is 1.23 ± 0.07, whereas for $NO_3$ steady state lifetimes of > 1500 s we derive a $[NO_3]_{DOAS,4}$ / $[NO_3]_{DOAS,1}$

| |
|---|
| ratio of 0.95 ± 0.05. |
| Page 16: The last paragraph in section 3.3.6 seems to be better placed in the previous section on the discussion of the residual layer and not in the comparison of CRDS and LP-DOAS. |
| This paragraph has been moved as suggested. |
| Generally, some material from the supplementary material would have been helpful in the main text; notably Fig S1, S2, S4 and S8. S1 and S2 are important to get an idea of the topography in the vicinity of the measurement site. Especially in the context of section 3.3.5 and 3.3.6 these figures are helpful. |
| Figures S2 and S8 have been moved to the main manuscript. |
| In many places mixing ratios are not expressed by volume. Examples are: P5,L23,L26,L27; P8, L15;P14,L19; P16,L4,L11. |
| Corrected |
| Page 3, Line 23: State the month(s) in 2011 explicitly here. |
| Done |
| Page 4, Line 13: "Fig. S1" should be highlighted. |
| Figure number highlighting will disappear in the final version |
| Page 6, Line 7: "broadbanding"? Is that the term to use? Moreover, the information in this line is redundant and was mentioned before on page 3. |
| Broadband is the correct term. This has been amended. |
| Page 9, L26: In Eq 5 the minus signs in the denominator are hardly discernible |
| We shall make sure that this is more legible in the type-set version. |
| Page 11, Line 6: IUPAC should get a reference. |
| IUPAC now referenced (Ammann et al., 2012) |
| Page 11, Line 29: "Fig. 7" should be highlighted. |
| Figure number highlighting will disappear in the final version |
| Fig. 1 & 2: There is a discrepancy on the third cold front arrival time (third blue vertical line). It may be better to merge Figs. 1 & 2 into one figure with six panels. If this could be done without loss of plot quality, this would help the reader. |
| The cold-front was on the 5$^{th}$ Sept. Figure corrected. We prefer to keep the Figures separate, but will make sure that they are located on the same page in final form. |
| Fig. 5 Units of seconds are missing on the vertical axis of the upper panel. |
| Corrected |
| Fig 6. The red and black trace refer to the left axis and radiation (in yellow) to the right axis. This is confusing, as the other panels are colour coded. |
| Figure modified |
| Fig 7. The lower panel does not seem to show LSUM, LNO. The stated time grid appears inconsistent with the dots in the figure (e.g. 30 min for LVOCS). |
| As stated in the text, NO mixing ratios were below the detection limit on the night of 30.08. Therefore LNO was not calculated and LSUM is not required. The sampling time of the VOC measurements was 20 mins (GC-MS) or 35 mins (GC-FID). The time resolution is however about 1 hour. This is now clearer in section 2.2.2. |
| The time resolution was ~ 1 hour (35 min sampling time), detection limits were around 1 pptv with an uncertainty of 10-15 %. |
| Page 22, Lines 2&3: ". . .of 4 the. . ." and "...at the 5 Großer Feldberg."? |
| Typos corrected |